# OFFLINE RL FOR NATURAL LANGUAGE GENERATION WITH IMPLICIT LANGUAGE Q LEARNING

**Charlie Snell, Ilya Kostrikov, Yi Su, Mengjiao Yang, Sergey Levine**
UC Berkeley; {csnell22,kostrikov,suyi,sherryy,svlevine}@berkeley.edu}

## ABSTRACT

Large language models distill broad knowledge from text corpora. However, they can be inconsistent when it comes to completing user specified tasks. This issue can be addressed by finetuning such models via supervised learning on curated datasets, or via reinforcement learning. In this work, we propose a novel offline RL method, implicit language Q-learning (ILQL), designed for use on language models, that combines both the flexible utility maximization framework of RL algorithms with the ability of supervised learning to leverage previously collected data, as well as its simplicity and stability. Our method employs a combination of value conservatism alongside an implicit dataset support constraint in learning value functions, which are then used to guide language model generations towards maximizing user-specified utility functions. In addition to empirically validating ILQL, we present a detailed empirical analysis of situations where offline RL can be useful in natural language generation settings, demonstrating how it can be a more effective utility optimizer than prior approaches for end-to-end dialogue, and how it can effectively optimize high variance reward functions based on subjective judgement, such as whether to label a comment as toxic or not[1].

## 1 INTRODUCTION

Large language models can acquire a remarkable amount of knowledge from large text corpora, and can be applied to a wide range of language-based tasks. However, such models are not designed to optimize any user-specified utility, instead requiring considerable trial and error to design prompts that coerce the models into producing desirable outputs (Liu et al., 2021; Brown et al., 2020; Min et al., 2021). In essence, standard unsupervised language model training only solves part of the problem, being effective at distilling down knowledge in large corpora, but relatively clumsy when applying this knowledge to solve user-specified tasks.

Reinforcement learning (RL) in principle can provide an effective framework for steering language models toward user specified tasks as long as the task can be represented by some utility

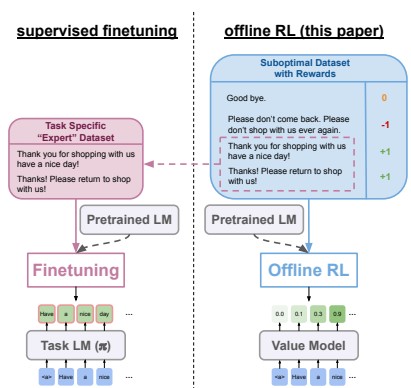

Figure 1: Offline RL differs from supervised learning in that it learns to maximize user-specified rewards from suboptimal data with reward labels.

function (i.e., a reward function); however, as outlined in Figure 2 contemporary methods suffer from high systems complexity and can require expensive human interaction. We need several conditions to make RL practical: (1) **Easy to use**: the underlying learning algorithm and workflow should be simple, stable, and scalable; (2) **Able to optimize user specified rewards**: the algorithm should be able to steer a language model toward maximizing any user-defined reward signal, from high-level task goals (e.g., book a flight) to low-level linguistic subtleties (e.g., avoiding rude or toxic speech); (3) **Practical for interactive applications**: the system should be able to handle a variety of tasks,

---

[1]Code at https://sea-snell.github.io/ILQL_site/

from generating text with desired properties to sequential turn-taking in settings such as dialogue tasks; (4) **Able to leverage existing data**: such a system should be able to directly utilize the large quantities of existing data, avoiding expensive and time-consuming online human interactions; (5) **Temporally compositional** (Emmons et al., 2021; Rafols et al., 2005): the method should be able to attain significant improvement over the average behavior in the data – not merely copying the best behaviors in the dataset, but actually distilling out underlying patterns in the relationship between rewards, task dynamics, and language to produce near optimal generations, even when the dataset demonstrates only mediocre task performance.

Offline RL provides a learning paradigm (Figure 1) that combines both supervised learning's ability to leverage existing data (criteria 4) with RL's ability to optimize arbitrary rewards and leverage temporal compositionality (criteria 2, 3, 5) (Levine et al., 2020; Kostrikov et al., 2021; Kumar et al., 2020; Janner et al., 2021; Chen et al., 2021; Yu et al., 2020; Kidambi et al., 2020). However, prior offline RL approaches for

| Method /
Criteria | Easy to Use | Able to
Optimize User
Specified
Rewards | Practical in
Interactive
Settings | Able to
Leverage
Existing Data | Temporally
Compositional |
|---|---|---|---|---|---|
| Supervised
Learning (BC) | ✔ | ✘ | ✔ | ✔ | ✘ |
| Filtered Fine
Tuning (%BC) | ✔ | ◯ | ✔ | ✔ | ✘ |
| Online RL | ✘ | ✔ | ✘ | ✘ | ✔ |
| ILQL (ours) | ✔ | ✔ | ✔ | ✔ | ✔ |

Figure 2: ILQL meets each of the five criteria for practical NLP RL methods we outline in Section 1.

language tasks are either based on dynamic programming, which enjoy the temporal compositionality (see Appendix A.2) but suffer from high systems complexity, hyper-parameter instability, and slow training times (Verma et al., 2022; Jaques et al., 2020; 2017) (meets criteria 5, fails 1), or methods based on conditional imitation or dataset value learning that are simple and stable to train, but do not provide the temporal compositionality enjoyed by "full" RL methods (meets criteria 1, fails 5) (Chen et al., 2021; Snell et al., 2022; Holtzman et al., 2018; Yang & Klein, 2021; Li et al., 2017; Krause et al., 2021). Motivated by all these criteria, we design a novel offline RL method based on dynamic programming with an implicit dataset support constraint (Kostrikov et al., 2021) that enjoys greater stability, fewer training-time dependencies (such as relying on approximate likelihoods from an external language model during training), and a more flexible decoding process than prior approaches (see Sections 4 and 6.4). Our method, ILQL, fine-tunes a transformer language model to predict the state-action $Q$ function and the state value function $V$ at each token. During training we perform iterative policy improvement by fitting a value function to an upper-expectile of the Q function, enabling us to learn policies that leverage temporal compositionality, significantly outperforming the data, while avoiding the need to execute expensive training-time procedures, such as sampling counterfactual utterances from the language model (Verma et al., 2022) (see Sections 5 and 6). Then at inference time we can simply steer a standard language model towards utility maximizing behavior, by perturbing the predicted likelihoods with our learned values functions (see Figure 3).

Our main contribution is twofold: (1) a novel offline RL algorithm, ILQL, for language models, that employs a stable optimization process that can flexibly learn high-performing policies from sub-optimal data in arbitrary sequential decision making settings, thus meeting each of the conditions laid out above; and (2) a detailed empirical analysis, not only demonstrating ILQL's ability to more consistently and stably adapt to many different utility functions than prior approaches, but also ILQL's unique ability to optimize stochastic or subjective reward functions, and its ability to discover optimal behaviors in the face of sub-optimal or unusual data distributions. In particular, in the controlled generation setting of generating non-toxic text, we demonstrate that ILQL, trained on both toxic and non-toxic comments, learns to produce fewer toxic outputs than the more standard approach of performing standard supervised fine-tuning on only non-toxic comments.

## 2  RELATED WORK

A number of prior works have explored combining online RL methods with language models for natural language tasks such as machine translation or summarization (Ranzato et al., 2015; Wu et al., 2016; Paulus et al., 2017; Wu & Hu, 2018). These works have demonstrated that RL can be an effective tool for steering language models towards satisfying utility functions. However, when it comes to settings that require multiple steps of human interaction, e.g., dialogue, these methods can quickly become impractical (Verma et al., 2022; Ghasemipour et al., 2020).

Offline RL addresses this shortcoming by removing all need for environment interaction or user simulators, instead operating purely on static datasets of prior human interaction. Several prior works have applied offline RL to NLP and more broadly sequence generation problems (Jaques et al., 2020; Verma et al., 2022; Jaques et al., 2017; Snell et al., 2022; Janner et al., 2021; Chen

et al., 2021). The most closely related to our work are those methods based on approximate dynamic programming (Jaques et al., 2020; 2017; Verma et al., 2022; Jang et al., 2022). While all these works present promising offline RL methods for NLP tasks, none of them provide a method that achieves the simplicity, stability, and ease-of-use aspect at the level of supervised learning. For example, Verma et al. and Jang et al. (Verma et al., 2022; Jang et al., 2022) define their action space at the "per-utterance" level (Verma et al., 2022), resulting in expensive decoding processes during training (Bender et al., 2021); and while Jaques et al. (Jaques et al., 2020; 2017) remove this issue by defining actions at the "per-token" level, the offline RL algorithm proposed requires querying likelihoods from a language model at RL training time, which adds an additional compounding source of approximation error and increases systems complexity at training time. Our proposed method instead operates both at the "per-token" level and trains in a fully self-contained way, without the need to simulate generation at training time or query likelihoods from a separate language model. This is achieved by combining an implicit dataset support constraint (Kostrikov et al., 2021) with a novel policy extraction method that takes advantage of the discrete "per-token" action space. The result of these design choices is a simple, stable, and effective method that is easy for NLP practitioners to pick up and apply to a variety of language-based tasks. In Section 6.4 we demonstrate our method's effectiveness in meeting these criteria through a series of ablations and comparisons.

Much prior work on steering language models towards desired behavior has done so without an explicit utility function, instead focusing on curating finetuning datasets (Zhang et al., 2018; Zellers et al., 2019; Rajpurkar et al., 2018). A more closely related line of work uses classifiers to guide LMs towards generating desired textual attributes (Yang & Klein, 2021; Ghazvininejad et al., 2017; Holtzman et al., 2018; Li et al., 2017). These methods are closely related to the prior work on offline RL. In RL parlance, such methods could be considered "policy extraction" methods with Monte Carlo value estimates. This can be interpreted as taking a single step of policy improvement which, though often effective (Brandfonbrener et al., 2021), is known to be suboptimal as compared to full dynamic programming methods (i.e., full Q-learning or actor-critic) (Kostrikov et al., 2021). We will demonstrate empirically in Section 5 that our offline RL method can lead to significant improvements in final performance as compared to such "single step" approaches, particularly when the training data is highly suboptimal for the desired task.

## 3 PRELIMINARIES: LANGUAGE GENERATION AS A REINFORCEMENT LEARNING TASK

**Token-level POMDP.** In this work, we formalize language generation tasks as a partially observable Markov decision process (POMDP). We define the POMDP $\mathcal{M}$ at the token level with $\mathcal{M} = (\mathcal{S}, \mathcal{A}, \mathcal{O}, \mathcal{T}, \mathcal{Z}, \mu_0, \mathcal{R}, \gamma)$. We define the agent's observation $h_t \in \mathcal{O}$ as a history of tokens with $h_t = \{t_0, t_1, t_2, t_3, ... t_{t-1}\}$; the action space $a_t = t_t \in \mathcal{A}$ is the set of possible next-tokens in our vocabulary which includes the special end-of-turn token $a_{\text{end}}$ (see Figure 3).

**Value-based offline RL.** In offline RL, the goal is to learn the optimal policy $\pi$ that achieves highest discounted cumulative reward from a static dataset $\mathcal{D}$ that was produced by some potentially suboptimal behavior policy $\pi_\beta$. In this work, we build on the implicit Q-learning (IQL) algorithm (Kostrikov et al., 2021), which approximates the Bellman optimality equation constrained to in-dataset actions

$$Q^*(s, a) = R(s, a) + \gamma \max_{a', \text{s.t. } \pi_\beta(a'|s') > 0} Q^*(s', a').$$

Instead of directly implementing the support constraint, IQL approximates the maximization on the right-hand side of the constrained Bellman operator with expectile regression:

$$L_V(\psi) = \mathbf{E}_{(s,a) \sim D}[L_2^\tau(Q_{\hat{\theta}}(s, a) - V_\psi(s))] \tag{1}$$

where $L_2^\tau(u) = |\tau - \mathbb{1}(u < 0)|u^2$. Increasing the hyperparameter $\tau$, more closely approximates the maximum. Then this approximation can be used to estimate TD-targets for the Q-networks:

$$L_Q(\theta) = \mathbf{E}_{(s,a,s') \sim D}[(R(s, a) + \gamma V_\psi(s') - Q_\theta(s, a))^2]. \tag{2}$$

IQL was designed for fully observable MDPs. However, in Section 4.1, we discuss how we adapt this formulation to the POMDP setting described above using sequence models.

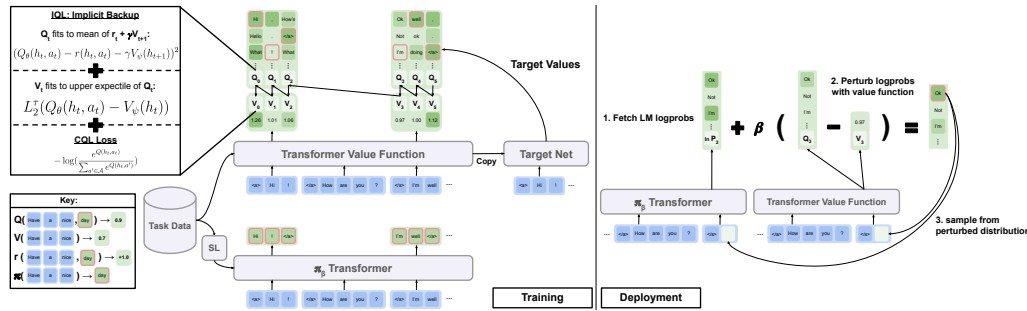

Figure 3: A diagram of our Implicit Language Q Learning algorithm. Left: ILQL training involves three transformers, each of which is finetuned from a standard pretrained model: (1) A $\pi_\beta$ model, finetuned with standard supervised learning. (2) A value function model, with Q and V on two separate heads; the value functions are trained with Bellman backups using a combination of conservatism and and an implicit dataset support constraint. (3) A target value network, which is a Polyak moving average of (2). Right: At inference time, we use our learned value functions to perturb the log probabilities of $\pi_\beta$ towards utility maximizing behavior.

**Supervised learning baselines.**    To align with NLP terminology, we refer to %BC, or supervised learning on curated or filtered data, as "filtered fine-tuning", and we refer to BC, or finetuning on unfiltered data as "fine-tuning". See Appendix A.4 for filtering details.

## 4  IMPLICIT LANGUAGE Q-LEARNING

Our main technical contribution implicit language Q-learning (ILQL), an offline RL algorithm for NLP tasks. ILQL is specifically designed to enable simple and efficient training of language models with user-specified reward functions, with a workflow that is similar to standard supervised learning.

ILQL builds on the IQL algorithm, extending it to the token-level POMDP that defines NLP tasks via the following modifications: (i) it integrates with sequence models to handle partially observable language generation tasks (Section 4.2); (ii) it utilizes a novel policy extraction method that directly perturbs the behavior policy $\pi_\beta$ with our learned value functions, rather than training a separate actor $\pi$, significantly improving performance and stability on NLP tasks (Section 4.1) and (iii) it adds a conservatism loss term (Kumar et al., 2020) to the Q-function, fixing a calibration issue in the policy extraction step. Figure 3 provides an overview of our method.

### 4.1  ADAPTING IMPLICIT Q-LEARNING TO LANGUAGE MODELS

**Implicit value function learning.**    Like IQL, our method learns both a value function and a Q-function, which bootstrap off each other through Bellman backups with *implicit* maximization through an expectile loss. This recursive process of fitting Q and V corresponds to iterative policy improvement subject to an implicit dataset support constraint, specified by the expectile used to fit V. Due to parameter sharing, we combine Eqn. 1 and 2 into a single loss function:

$$L_{Q,V}(\theta) = \mathbf{E}_{\tau \sim D} \left[ \sum_{i=0}^{T} (R(h_i, a_i) + \gamma V_\theta(h_{i+1}) - Q_\theta(h_i, a_i))^2 + L_2^\tau(Q_{\hat\theta}(h_i, a_i) - V_\theta(h_i)) \right]$$

In contrast to IQL, we sample sequences of tokens instead of individual transitions to handle partial observability, such that for each time step, the values are predicted based on a full history.

**Policy extraction.**    IQL (Kostrikov et al., 2021) uses AWR policy extraction (Peng et al., 2019), which distills the Q-function into a policy with a $e^{\beta(\hat{Q}-V)}$ weighted log-likelihood loss. However, as we discuss in Section 6, we found this somewhat unstable to train on language models, likely due to the high-variance gradients induced by the advantage weights. Fortunately, the value learning procedure in IQL is independent of policy extraction, so instead of attempting to train a model to represent the optimal policy, we use the learned Q and V values to directly perturb samples from a model finetuned via supervised learning to model $\pi_\beta$ (see Figure 3). To this end, we compute a modified likelihood for each token by adding its advantage $Q(h, a) - V(h)$ to its logits under the $\pi_\beta$ model, with a multiplier $\beta$. We can then renormalize these pseudo-logits and sample them, resulting in

the implicit policy $\pi(a|h) \propto \pi_\beta(a|h)e^{\beta(Q(h,a)-V(h))} = \exp(\log(\pi_\beta(a|h)) + \beta(Q(h,a) - V(h)))$. This does not require training a separate actor, only a behavioral model $\pi_\beta$, which can be trained with the standard and stable supervised finetuning objective.

However, naïvely performing policy extraction in this way can perform poorly due to over-smoothed probabilities in $\pi_\beta$ that may be nonzero for extremely unlikely tokens. In this case, samples from $\pi_\beta$ may be out of distribution for $Q$ and $V$, and might have erroneous values. To fix this calibration issue, we can either further decrease the probability of low probability actions in $\pi_\beta$ by performing top-p filtering or tuning a temperature parameter, or we can explicitly push down OOD Q-values during training. We implement the latter by adding a small amount of NLL loss to the Q values, which corresponds to the additional loss terms introduced by CQL (Kumar et al., 2020) with a uniform KL regularizer. Since ILQL actions are discrete tokens, as opposed to the original CQL method (Kostrikov et al., 2021), which operates on continuous action spaces, this CQL loss term is no more expensive than, and in fact equivalent to, a standard cross-entropy loss at the token level. We find that both of these approaches often work in practice, but prefer the latter, finding that it requires less tuning for policy extraction at inference time. Our full loss function is therefore:

$$L_{Q,V}^c(\theta) = L_{Q,V}(\theta) - \alpha \mathbf{E}_{\tau \sim D} \log \left( \frac{e^{Q_\theta(s_i,a_i)}}{\sum_{a' \in \mathcal{A}} e^{Q_\theta(s_i,a')}} \right)$$

In early experiments, we found that decoding using the CQL regularized value functions alone, without $\pi_\beta$, required careful tuning of the CQL weight $\alpha$. When the CQL regularized value function is combined with $\pi_\beta$ for policy extraction, it mitigates this issue with hyper parameter sensitivity, and simply setting the CQL weight $\alpha$ to an arbitrary small value less than 1 typically works well.

## 4.2 Architectures for Implicit Language Q-Learning

We use GPT-2 small as the base model for all transformers in our experiments. Our value function transformer has three MLP heads: two independently initialized and trained Q heads and one V head. Each head has two layers, with a hidden dimension twice that of the embedding dimension. Our target Q value is parameterized as the minimum prediction of both Polyak averaged target Q heads: $\hat{Q} = \min(Q_1, Q_2)$ (Fujimoto et al., 2018). As in standard language modeling, the transformer's causal masking enables us to perform Bellman updates over entire sequences in parallel.

## 5 Proof of Concept: Multi-Step Offline RL on Wordle

The lack of configurable task settings and reliable evaluations has arguably slowed down progress in applying sophisticated RL algorithms to language and dialogue tasks (Jiang et al., 2021; Deriu et al., 2021; Curry et al., 2017). To address this, we present the Wordle game (Lokshtanov & Subercaseaux, 2022) as an easy-to-use but challenging benchmark task to test the capabilities of offline RL algorithms. In this section, we use this task to construct situations where we would expect offline RL to lead to significant improvement over simpler methods based on supervised learning or single-step improvement (e.g., single-step RL-style or filtered supervised learning methods).

**Multi-step RL: a motivating example.** General value-based RL methods based on solving the Bellman equation described in Section 3 can be viewed as iteratively improving the policy: each update sets the current value $Q(h_t, a_t)$ to be the reward plus the *maximum possible* next time step value according to the current value function. This is in contrast to "single-step" update methods, which do not *recursively* update the value function, instead only learning to estimate the value of the dataset and then greedily selecting the maximal-value action during inference. Classic examples of such methods use Monte Carlo regression or single-step RL (Sutton et al., 1998) to train the value function, and then greedily choose actions at test-time, though a number of different methods of this sort have been proposed for guided language generation in the literature (Yang & Klein, 2021; Ghazvininejad et al., 2017; Holtzman et al., 2018; Li et al., 2017). Such methods have been referred to in the NLP literature as "reward models" (Young et al., 2017; Gu et al., 2016; Su et al., 2016) and in the offline RL literature "one step RL" or SARSA (Brandfonbrener et al., 2021). Some works also proposed behavioral cloning methods that filter the training data or use conditioning to clone high-reward trajectories (Chen et al., 2021) – though these methods are based on different principles, they also employ Monte Carlo estimates of the cumulative reward in place of value functions learned with dynamic programming. While in principle such methods should not lead to optimal policies, in practice they often constitute an appealing approximation due to their ease of use.

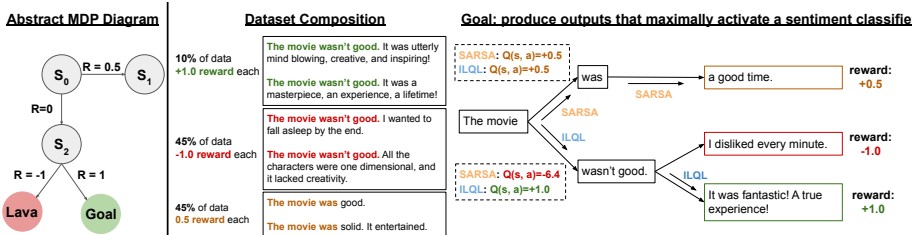

Figure 4: Left: an abstract depiction of an MDP where single-step RL fails to discover the optimal policy. Right: A notional illustrative example where we might expect full "multi-step" RL methods (such as ILQL) to perform significantly better than "single-step" methods. In this example, good utterances tend to start with "The movie was...", while bad utterances start with "The movie wasn't..." However, the *very best* examples also start with "The movie wasn't...", requiring multi-step planning or multiple steps of policy improvement to derive effective strategies. Methods that implement just a single step of policy improvement will fail to produce maximally positive sentiment outputs. While this example may appear somewhat contrived, we see in our experiments that multi-step RL methods do lead to improvements in a number of more real settings.

Since ILQL performs multiple steps of policy improvement, it can significantly improve over Monte Carlo estimators or single-step RL when the underlying data is sub-optimal. One example corresponds to the notional task in Figure 4, in which the optimal sequence of actions requires traversing a state that's also frequented by sub-optimal examples. In this case, single-step RL will learn to take actions that appear safer according to the dataset — such as the transition "The movie" $\rightarrow$ "was" in Figure 4 — whereas full ("multi-step") RL methods would recover the optimal policy. We demonstrate this empirically on the Wordle game below.

**Wordle dataset.** Our Wordle task is designed to allow us to use data from real humans in a sequential decision-making setting while still enabling objective simulated evaluation and the flexibility to compose datasets with different properties, thus providing an effective benchmark for validating a variety of approaches. Wordle is a word guessing game in which the agent gets 6 attempts to guess a certain word, and at each turn receives feedback from the environment about which letters from the guessed word are and are not in the true word (see Appendix A.5 for more details). While Wordle may appear distinct from many natural language tasks, it shares a number of high-level properties, such as non-deterministic dynamics and a sequential turn-based structure, with more complex language domains like dialogue, making it well suited for a first evaluation of NLP-focused RL methods.

**Synthetic Wordle task.** We constructed a synthetic Wordle dataset to serve as a benchmark that is specifically intended to evaluate how well a particular method can perform multiple steps of policy improvement. This task intended to specifically bring out the failure mode of single step methods in a setting suitable for testing offline RL methods with sequence models. The dataset consists of data sampled from three behavior policies, each corresponding to one of the branches in Figure 4: (1) $\pi_{\text{upper bound}}$, a high-performing policy, corresponding to the path $S_0 \rightarrow$ Goal. (2) $\pi_{\text{adversarial}}$, which behaves the same as $\pi_{\text{upper bound}}$ for the first two actions ($S_0 \rightarrow S_1$) and then behaves suboptimally ($S_0 \rightarrow$ Lava). (3) $\pi_{\text{suboptimal}}$, a policy of moderate performance, corresponding to $S_0 \rightarrow S_1$. Measuring the predicted Q values from our models trained on this distribution, in Figure 6 (right) we observe that ILQL assigns higher values to actions corresponding to the paths towards "misleading states" (i.e. $S_2$) than those to the "goal states" (i.e. $S_1$), whereas single-step RL shows the exact opposite preference, confirming both our hypothesis that this type of MDP would be amenable to multiple steps of policy improvement, and that ILQL as an algorithm is able to perform such policy improvement. Of course, the structure of real-world NLP tasks might not necessarily reflect this setting – as we show in the next section, ILQL still often attains improvement over one-step and BC-based methods, though it is more difficult to discern the particular structure that makes this possible in more realistic tasks.

**Validating on natural Wordle data.** While the synthetic setting explored above was specifically designed to demonstrate a dramatic difference between ILQL and single-step RL, the findings still transfer to more realistic settings. In Table 6, we demonstrate ILQL outperforming single-step RL on a natural dataset of Wordle games scraped from Twitter (see Appendix A.5 for details).

## 6 NATURAL LANGUAGE EXPERIMENTS

Next, we evaluate ILQL on two realistic language tasks. We first identify scenarios in which one might expect offline RL to be particularly beneficial: (1) tasks that demand repeated interactions,

| method | standard | y/n | cons. y/n |
|---|---|---|---|
| ILQL | -5.21 ± 0.13 | **-5.57**±0.13 | **-6.57** ± 0.18 |
| 1-step RL | -5.14 ± 0.13 | -5.91 ± 0.14 | -7.63 ± 0.20 |
| Filtered FT | -5.07 ± 0.13 | -7.48 ± 0.21 | -9.13 ± 0.22 |
| FT | -5.25 ± 0.13 | -10.85 ± 0.27 | -15.16 ± 0.35 |

| train/eval | standard | y/n | cons. y/n |
|---|---|---|---|
| standard | **-5.21** ± 0.13 | -11.12 ± 0.30 | -14.97 ± 0.36 |
| y/n | -5.41 ± 0.12 | **-5.57** ± 0.13 | -8.24 ± 0.22 |
| cons. y/n | -5.29 ± 0.13 | **-5.42** ± 0.13 | **-6.57** ± 0.18 |

Table 1: Left: comparing ILQL to baselines on our Visual Dialogue rewards. "Cons. y/n" refers to the "Conservative y/n" reward. ILQL successfully optimizes for many different rewards, even those for which the data is sub-optimal (e.g. BC performance). Right: Evaluating each ILQL agent on other rewards. Agents generally perform worse on rewards for which they were not trained.

like dialogue; (2) data that is highly sub-optimal under its utility function; (3) settings with highly stochastic rewards based on subjective human judgement (e.g., avoiding toxic language). Taking into account these three scenarios, we evaluate ILQL on (1) a goal-directed question asking task based on Visual Dialogue (Das et al., 2016), where achieving high rewards on a diverse set of metrics during repeated interactions is desirable, and (2) a Reddit comments generation task with highly subjective and noisy reward functions (toxicity ratings or upvotes). For general experiment details see Appendix A.4.

### 6.1 EVALUATING DIVERSE REWARDS ON VISUAL DIALOGUE

**Visual Dialogue dataset.** We use the Visual Dialogue dataset (Das et al., 2016) to evaluate our algorithm's ability to optimize many different reward functions in complex dialogue settings. The task involves both a question asking and question answering agent, the latter of which is presented with an image and tasked with answering the former's questions about the image. Instead of using this task as a question answering task, we follow Das et al. (Das et al., 2017) and train our agents to ask questions, with rewards based on how well the ground-truth image can be predicted from the resulting dialogue. For evaluation, we use the model from Das et al. (Das et al., 2017) as our environment simulator. To allow our agents to operate entirely in the space of natural language, we treat the image embedding as part of the reward function, using the supervised model proposed by Das et al. (Das et al., 2017) to predict the image embedding from the dialogue. See Figure 5 for example dialogues in this domain. We chose this environment specifically because (1) it has been previously studied in the context of RL (Das et al., 2017); (2) as a dialogue game, automated evaluation is more reliable than other tasks; and (3) the Q&A structure enables some temporal compositionality (i.e., the answer to one question may prompt new more specific questions).

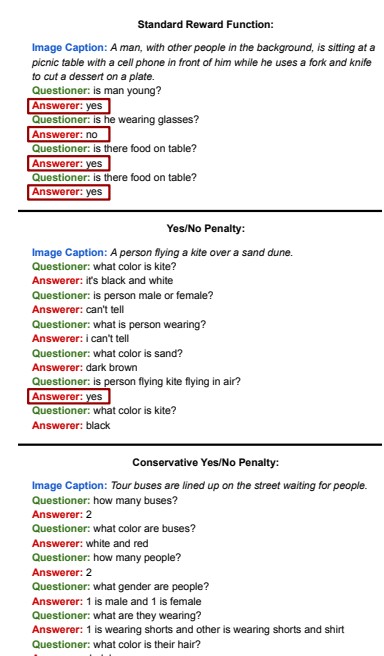

Figure 5: Dialogues from different agents on the Visual Dialogue task. We observe qualitative differences depending on the agent's reward. The "standard" agent asks many yes/no question, whereas adding a string-match penalty for yes/no questions prevents many of such questions from being asked, and adding a more conservative yes/no penalty prevents all of such questions.

**Visual Dialogue task.** The agent receives a reward of -1 for each turn in which the ground truth image can't be predicted accurately from the dialogue, otherwise the agent receives a reward of 0 and the environment ends interaction. For details on the task setup, see Appendix A.6. Since the Visual-Dialogue dataset was largely designed for supervised learning agents, the data is already near optimal for the original task. However, if we shift the reward function such that the data is no longer optimal, we can observe large gains from offline RL. We therefore use this domain to demonstrate offline RL's flexibility to adapt to different reward functions. We consider three rewards: "standard", "y/n", and "conservative y/n". "Standard" is simply the reward described above and detailed in Appendix A.6. "y/n" adds to the "standard" reward a penalty for asking questions that produce yes or no answers, by assigning a reward of -2 each time the *other* speaker says "yes" or "no". This is challenging, because while the data contains many yes/no questions, the goal is not for the agent *itself* to avoid those words, but rather

| method | toxicity | upvotes real | upvotes model |
|--------|----------|--------------|---------------|
| ILQL | **0.0**±0.0 | **9.83**±0.04 | **10.0**±0.0 |
| single-step RL | **0.0**±0.0 | 6.23±0.15 | **10.0**±0.0 |
| Filtered FT | -0.74±0.07 | 7.06±0.14 | 7.86±0.13 |
| FT | -3.51±0.13 | 4.87±0.16 | 4.87±0.16 |

| train/eval | toxicity | upvotes model |
|------------|----------|---------------|
| toxicity | **0.0**±0.0 | 9.07±0.09 |
| upvotes gold | -5.00±0.00 | 9.83±0.04 |
| upvotes model | -5.00±0.00 | **10.0**±0.0 |

Table 2: Left: A comparison of ILQL against baselines on our Reddit comment rewards. ILQL never generates undesirable comments on 2 out of 3 rewards, whereas fine-tuning on filtered data occasionally does. Right: Evaluating each Reddit ILQL agent on other rewards. Agents trained on one reward are less optimal on others.

avoid utterances that cause the *other* speaker to use them. Not all simple questions produce literal "yes/no" answers, so our third reward further penalizes brief responses, such as "I can't tell", "no it isn't", "yes it is", "I don't know". This reward function assigns a -2 reward to a set of low-information responses using a handful of conservative string matching heuristics, detailed in Appendix A.6.

**Results on optimizing diverse rewards**    We demonstrate that ILQL learns a policy distinct from the dataset behavior policy and can optimize many different rewards in Table 1 left, where we can see that ILQL is able to outperform baselines on most of our Visual Dialogue reward functions. Agents optimized for each reward learn different behaviors as well: in Table 1 right, we see that offline RL agents trained on one reward function are generally suboptimal on others. Figure 5 demonstrates this qualitatively: without the "yes/no" penalty, our policies ask many "yes/no" questions, but under its presence policies tend to ask other questions instead. Even when the underlying data is highly sub-optimal for a given reward function, ILQL is able to determine the desired behavior. In addition to our reward-based evaluations, we also provide language quality evaluations for all baselines on each reward function in Section A.11.

## 6.2 Subjective Rewards on Reddit Comments

**Reddit comments dataset.**    To evaluate our agents on minimally curated and maximally diverse open-domain text with highly stochastic reward functions based on subjective human judgement, we train ILQL on a large dataset of 4 million Reddit comments from [2]; our agents are given a parent comment or post as context and then trained to produce replies that satisfy one of two rewards: "toxicity" and "upvotes real". Given that this is internet text, the data contains toxic language, so for our "toxicity" reward, we train our agents to satisfy a toxicity filter, which gives rewards -10, -5, and 0 for toxic comments, moderately toxic, and non-toxic comments respectively. Our second reward function incentivizes comments that would receive a positive number of upvotes, rewarding +10 for positive upvotes and 0 negative. We automatically evaluate our upvote agents with a finetuned RoBERTa-base model, that predicts whether a comment will receive positive upvotes. We train this model on a held-out split of the data (see Appendix A.7 for more details). We train agents on both the ground truth upvotes (denoted "upvotes real") and on this upvote model's predicted reward (denoted "upvotes model").

**Results on optimizing noisy rewards.**    In natural language tasks, we may need to optimize stochastic, high-variance reward functions based on subjective judgement, such as whether a Reddit comment should be flagged as toxic. Such stochastic settings should be expected when multiple users with differing opinions provide reward labels. Offline RL, by design, is robust to environment stochasticity, and therefore should be able to optimize such noisy environments. We use the Reddit toxicity and upvote tasks to study how well ILQL can handle such settings. As we can see in Table 2 top, ILQL is surprisingly able to get a perfect or near-perfect score on these more subjective settings, whereas more standard approaches, such as filtered finetuning on only non-toxic or only positive upvote comments, perform significantly worse (i.e. generates more comments flagged as toxic or predicted to have negative upvotes). Additionally, we see in Table 2 bottom that agents trained on one reward function are generally less optimal for others, confirming that ILQL is effectively specializing its behavior for each utility function. We have additional complementary experiments studying this effect in Appendix A.8. In addition to our reward-based evaluations, we also provide language quality evaluations of our agents in Section A.11, and a preliminary user study in Appendix A.12.

## 6.3 Choice of Offline RL algorithm

We compare ILQL to four other RL methods: a per-token version of CQL, an adaptation of the $\psi$-learning as proposed by Jaques et al. (Jaques et al., 2020; 2017), decision transformer (DT) (Chen et al., 2021), and single-step RL (Yang & Klein, 2021; Ghazvininejad et al., 2017; Holtzman et al., 2018; Li et al., 2017). In Table 3, we see that ILQL significantly outperforms baselines, and also has the second lowest hyper-parameter variance, just behind single-step RL, confirming our hypothesis that ILQL can provide both high relative performance and training stability.

---

[2]https://www.kaggle.com/code/danofer/reddit-comments-scores-nlp/notebook

## 6.4 ILQL ABLATIONS

To understand which components of ILQL enable both good results and greater ease-of-use than prior offline RL approaches for language tasks, we abalate ILQL's main design decisions: the choice of a per-token action space, our value learning method, and our policy extraction strategy.

We evaluate these comparisons on the Visual Dialogue "yes/no" reward because (1) it is important to compare offline RL methods on a challenging and realistic sequential decision problem like dialogue, and (2) since the Visual Dialogue data is already "near-expert" for the "standard" reward, the "yes/no" reward is better able to differentiate between methods.

| method | max score | $\sigma$ w.r.t hparams |
|---|---|---|
| ILQL | **-5.57**±0.13 | 0.46 |
| CQL | -7.32±0.17 | 1.98 |
| $\psi$ | -10.05±0.18 | 0.60 |
| single-step RL | -5.91±0.14 | **0.35** |
| DT | -6.70±0.17 | 1.15 |
| Filtered Fine-tuning | -7.48±0.21 | 0.72 |
| Fine-tuning | -10.85±0.27 | - |

Table 3: Comparison of ILQL and other offline RL methods on the visual dialogue "y/n" task. ILQL performs better with lower hyperparameter sensitivity. Results show best hyperparameter settings for each method.

**Ablations on per-token vs. per-utterance actions.** We hypothesize that, since a per-token Q function enables an efficient search through the utterance action space, performing offline RL at the token level, rather than the utterance level, can yield cheaper inference and better performance.

We compare ILQL to: (1) ILQL (utterance): a per-utterance adaptation of ILQL that removes the conservatism loss term and performs Bellman backups at the utterance level instead of the token level; (2) single-step RL (utterance): a per-utterance version of single-step RL; and (3) CHAI (Verma et al., 2022): an adaptation of CQL for use on language models at the per-utterance action level. For policy extraction, each baseline uses EMAQ (Ghasemipour et al., 2020), where we sample $N$ utterances from a learned behavior policy and then re-rank with the Q function.

We see in Table 5 that ILQL outperforms per-utterance ILQL and single-step RL, while also running inference ∼4x faster on a single T4 GPU. When tuned well, we see that CHAI perform similarly to ILQL. However, CHAI is less stable with respect to hyperparameters and is >2x slower at inference time.

**Ablations on policy extraction strategies.** In adapting IQL (Kostrikov et al., 2021) to sequence models, we design a novel policy extraction strategy, as described in Section 4. We compare our novel extraction procedure to two baselines: (1) the standard AWR-based (Peng

| method | max score | $\sigma$ w.r.t hparams |
|---|---|---|
| ILQL | **-5.57**±0.13 | 0.46 |
| ILQL (AWR) | -5.96±0.13 | 2.82 |
| ILQL (GOLD) | -7.58±0.21 | 0.61 |

Table 4: Policy extraction techniques for ILQL.

et al., 2019) policy extraction method used in IQL (Kostrikov et al., 2021) and a number of other offline RL algorithms (Nair et al., 2020; Wang et al., 2020), and (2) GOLD (Pang & He, 2020) an off-policy gradient based method for natural language generation. We expect that our approach should generally yield better performance, while also being easier to tune than these baselines. We apply AWR and GOLD extraction to our best performing ILQL value function, denoting this as "ILQL (AWR)" and "ILQL (GOLD)". Table 4 demonstrates that ILQL is both more stable and better at extracting good performance from a given value function than the well established AWR-style extraction and GOLD. Additionally, the AWR and GOLD extraction methods require tuning additional hyper-parameters at training time rather than just at inference time, decreasing flexibility and increasing the time and effort spent tuning parameters and re-training.

## 7 CONCLUSION

We proposed ILQL, an offline RL method for steering language generation to fulfill a variety of desirable conversational behaviors. Through experiments ranging from word games and goal-directed question asking to optimizing upvotes and minimizing toxic language, ILQL shows that offline RL can serve as a strong alternative to the method landscape of language generation dominated by language model finetuning on manually filtered datasets and classifier guidance. We hope the positive results from ILQL will inspire more work on offline RL for dialogue, and lead to more controllable language models that directly optimize user-specified utility functions for a wide range of tasks in text generation. Lastly, we acknowledge that our method is generally more computationally expensive than more standard supervised learning approaches to language generation ( 2x more training time in our experiments), since it requires 3 separate transformer networks during training and 2 during inference. Future work should study alleviating this cost.

# 8 ETHICS STATEMENT

We acknowledge that any utility optimization method can be used to aid or harm, we hope these future works consider ethical uses of offline RL. Additionally, our method also has its limitations: for example, ILQL may not be effective when datasets are highly suboptimal. Offline RL would also not be ideal in settings which require distributional constraints, such as fairness. We hope that practitioners will take these limitations into account when applying our method.

# 9 REPRODUCIBILITY STATEMENT

To promote reproducibility, we present extensive results of all hyper-parameter settings we tried for all baselines in the appendix. We also describe all experimental details. Lastly, we have also attached in the supplemental source code for reproducing all the experiments in this submission.

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

## A  APPENDIX

### A.1  JUSTIFICATIONS FOR OFFLINE RL IN DIALOGUE

Dialogue tasks are one of the most rich and interactive settings in NLP, and as we will argue, these properties make it an ideal target for applying offline RL. RL in general presents an elegant and highly desirable utility optimization framework for sequential decision making settings, such as dialogue. However, solving realistic interactive tasks with online RL requires either repeated real-world interaction or building a realistic simulator of the environment. In the case of dialogue, such online interaction means communicating with real humans, which may be impractically expensive and time-consuming with contemporary sample-inefficient online RL methods (Schulman et al., 2017; Haarnoja et al., 2018), and building a realistic simulator of human responses may be largely intractable in sufficiently rich or complex dialogue settings. Offline RL, on the other hand, avoids both of these heavy requirements, by, just as many recent breakthroughs in the field of NLP (Brown et al., 2020; Radford et al., 2019; Devlin et al., 2018), operating purely on previously collected data, which is wildly available on the internet in general. Offline-RL therefore presents an ideal approach for flexibly steering language models towards the successful completion of dialogue tasks in a way that effectively leverages existing data, just as supervised learning does.

## A.2 TEMPORAL COMPOSITIONALITY

It has been well studied in the RL literature that value-function based RL methods are capabale of a form of temporal compositionality (Emmons et al., 2021; Rafols et al., 2005) or "stitching" in their utility optimization. Specifically this refers to an RL algorithm's ability to stitch together the locally optimal parts of suboptimal trajectories to find globally optimal behavior. The result of this stiching is a learning algorithm which can distill the optimal behavior out of a dataset that contains only suboptimal demonstrations.

## A.3 FULL TOKEN-LEVEL POMDP FORMULATION

We expand on the POMDP definition presented in Section 3. In order to apply RL to interactive language settings, we need to formalize dialogue generation and other NLP tasks as a partially observable Markov decision processes (POMDP). We define the POMDP $\mathcal{M}$ at the token level with $\mathcal{M} = (\mathcal{S}, \mathcal{A}, \mathcal{O}, \mathcal{T}, \mathcal{Z}, \mu_0, \mathcal{R}, \gamma)$. We define the agent's observation $h_t \in \mathcal{O}$ to be a history of tokens with $h_t = \{t_0, t_1, t_2, t_3, ...t_{t-1}\}$; the action space $a_t = t_t \in \mathcal{A}$ is defined to be the set of possible next-tokens in our vocabulary, which includes the special end-of-turn token $a_{\text{end}}$. The agent's policy then corresponds to a mapping $\pi : \mathcal{O} \to \mathcal{P}(\mathcal{A})$. Many tasks such as dialogue have an underlying state $s_t$ that goes beyond just the sequence history, which can encompass things like the speaker's mental state. The environment transitions $\mathcal{T}(\cdot|s_t, a_t)$ are defined as a function of this $s_t$. In particular, in domains, such as dialogue, the dynamics are trivial within the agent's utterance (the selected token is deterministically appended to the history), but when the policy outputs a special "end of turn" token, the other speaker gets a turn, which is subsequently appended to the history. In other tasks, where the goal is to generate a single utterance, such as generating a summary or a single Reddit comment, the episode ends when the policy produces the end token. The agent receives a reward, defined by $R(s_t, a_t) \to \mathcal{R}$, after each action taken. However, in all the settings we consider, the agent receives non-zero reward $r_t$ only after producing an "end of turn" token, rather than densely at every token in the agent's utterances.

While some prior works (Verma et al., 2022; Jang et al., 2022) have considered actions at the utterance level, defining decision processes at the token level can yield a more effective search over the exponentially large utterance action space, simply by selecting tokens with high estimated values. Typically, searching over a per-utterance action space requires a Monte Carlo process of sampling multiple full utterances and then re-ranking with estimated values, which can generally bring additional computational complexity at both training and inference time. In Section 6.4, we demonstrated the effectiveness of learning at the token level through an ablation study that compares with learning at the utterance level.

## A.4 GENERAL EXPERIMENT DETAILS

Here we outline architecture and hyper-parameter details of all our models and baselines.

**ILQL experiment details.** We run all of our experiments on GPT-2 small transformer architectures, with the supervised learning policy on one transformer and Q and V heads on a separate one. The target Q network is also on a separate transformer. In all our experiments we initialize with GPT-2 pre-trained weights, except in the case of Wordle, where we initialize randomly. Additionally, Wordle uses a different token set: the set of 26 characters, plus an additional token for each "color". We train all RL baselines with double-Q learning, using two separate heads on the same transformer model as the two Q-functions. Our target Q networks are Polyak-averaged with decay factor 0.005 for both the transformer and the Q function head. We use $\gamma = 0.99$ for all offline-RL experiments. All value function heads are two layer MLPs with hidden dimension twice that of the transformer's embedding dimension. Our MLPs used ReLU non-linearities and no dropout. We used the AdamW optimizer for all experiments, with a learning rate of 1e-4 on the Reddit and Visual Dialogue tasks and 1e-5 on the Wordle task. We used no weight decay in the training any of our models, and we used a dropout rate of 0.1 inside the transformer. We trained all Wordle models with a batch size of 1024, all Visual Dialogue models with a batch size of 64, and all Reddit models with a batch size of 32. We always truncate token sequences to length 1024, except on Reddit tasks, in which we truncate to length 512.

Except on the Reddit comment task, we train ILQL on each of $\tau = \{0.7, 0.8, 0.9\}$, and we also evaluate each on $\beta = \{4, 8, 16, \infty\}$. Where $\beta = \infty$ refers to acting greedily according to only

the Q function. On the Reddit comment tasks, we only train with $\tau = 0.6$ and evaluate on $\beta = \{1, 2, 4, 8, 16, 32\}$. On all tasks, we report the setting with the greatest performance.

For the NLL (CQL) loss term applied to ILQL, we used a weight $\alpha$ of 1.0 on all VisualDialogue experiments, 0.25 on all Reddit Comment experiments, and 0.0001 on Wordle. These values were tuned by hand. Generally, we find this loss parameter to not be too critical to performance; we tune it a little at first for each task and then don't worry about it.

All ILQL models and all baselines were trained on a single GPU until convergence. Training never exceeded three days (or 72 V100 hours).

**Evaluation details.** During evaluation, we use greedy decoding to generate utterances on all tasks and baselines, except the Reddit Comments tasks, where we sample instead. All experiments are evaluated on 1024 task-instances from an unseen evaluation set. We use our BC baseline model as $\pi_\beta$ for guiding ILQL's perturbation-based policy extraction.

**Fine-tuning (BC) baselines.** We train our fine-tuning baselines with the same optimization parameters (i.e., weight decay, dropout, learning rate, batch size) and initialization as ILQL. We use early stopping: when the validation loss exceeds the training loss, we stop training. Unlike our ILQL value function models, we use a linear head on top of the transformer to parameterize our BC policy, as is standard for language model finetuning. The only difference between our fine-tuning and standard language model training is that instead of finetuning the model to predict the whole sequence of states and actions, we only finetune the model to predict the agent's own actions or utterances.

**Filtered Fine-tuning (%BC) baselines.** For our filtered fine-tuning baselines, we train models on datasets filtered for the top reward trajectories. Specifically we filter for the top $\{10\%, 30\%, 50\%\}$ for Wordle, $\{10\%, 20\%, 30\%\}$ for Visual Dialogue, and for Reddit, since our rewards are discrete, we define filtered fine-tuning to mean just training on the data-points with the maximum reward label. For each task, we train models on each of these percentages and report the performance of the best setting found. We use the same hyper-parameters as our Fine-tuning (BC) baselines for training these models.

**Decision transformer baseline.** Our decision transformer baseline follows from Chen et al. (Chen et al., 2021), except we initialize with pretrained GPT2 weights. All hyperparameters are identical to those used to train our BC baselines. To evaluate decision transformer on our Visual Dialogue "y/n" reward, we swept over a broad range of conditional reward-to-go values: $\{-11, -10, -9, -8, -7, -6, -5, -4, -3, -2, -1, 0\}$. We report the setting with the best performance.

**single-step RL baselines.** single-step RL baselines are implemented as ILQL with $\tau = 0.5$ and all other hyper-parameters are identical to those used with ILQL as described above. Except on the Reddit comment tasks, we evaluate all single-step RL models on $\beta = \{4, 8, 16\}$, and report the setting with the greatest performance. On the Reddit comment tasks, we show the best performance from $\beta = \{1, 2, 4, 8, 16, 32\}$.

**Per-utterance ILQL.** For "ILQL (utterance)", we train models with $\tau = \{0.7, 0.8, 0.9\}$. We also evaluate each model with number of EMAQ-style (Ghasemipour et al., 2020) samples chosen from N=$\{4, 8, 16\}$. We report the setting with the best task performance. "single-step RL (utterance)" is a special case of "ILQL (utterance)" with $\tau = 0.5$, which we evaluate on each of N=$\{4, 8, 16\}$ and report the setting with the best performance. The architecture for "ILQL (utterance)" and "single-step RL (utterance)" is largely identical to that of per-token ILQL, with the main difference being that Bellman backups are performed at the utterance level instead of the token-level. As a result of this difference, Q-heads map to a scalar at the end of an utterance instead of a vector at every token with length equal to the size of the vocabulary. We include comparisons to Per-utterance ILQL in Table 5.

**CHAI baseline.** Our CHAI baseline is adopted from Verma et al. (Verma et al., 2022). The tasks we consider only require utterance actions; no auxiliary actions, like the price proposal action required by that of Verma et al. (Verma et al., 2022)'s bargaining task. We therefore only adopt the components from CHAI relevant to utterance level actions. In order to compute CHAI's CQL loss at the utterance

| method | max score | $\sigma$ w.r.t hparams | inference time per-dialogue (sec) |
|---|---|---|---|
| ILQL | **-5.57**±0.13 | 0.46 | **5.10**±0.12 |
| ILQL (utterance) | -5.89±0.14 | 0.51 | 22.1±0.47 |
| single-step RL (utterance) | -7.35 ± 0.17 | **0.21** | 20.38±0.41 |
| CHAI | **-5.57**±0.13 | 1.11 | 12.13 ± 0.25 |

Table 5: On the VisualDialogue "y/n" reward, we compare per-token ILQL to per-utterance ILQL, per-utterance single-step RL, and CHAI (Verma et al., 2022). We observe that per-token ILQL is generally much faster at inference time than per-utterance methods, while also outperforming or performing equivalently to all per-utterance baselines. All evaluations were performed on a single T4 GPU. All baseline implementations build on the same core code for sampling utterances, with a handful of method specific runtime optimizations in each case.

level, we need to sample counterfactual utterances for each action in the training data, which can be highly expensive and can greatly slow training. Following Verma et al. (Verma et al., 2022), we amortize this cost at the risk of inducing some bias by caching 5 counterfactual samples for each action in the training data as a preprocessing step. In our case, this preprocessing step took over 30 hours to execute on a V-100 GPU for the full the Visual Dialogue training set. As in all our other experiments, we train two Q networks (Fujimoto et al., 2018), where our target Q value is parameterized as the minimum of both Polyak averaged target Q heads. We train our CHAI models with a batch size of 16 and otherwise all other hyperparameters are identical to those used with ILQL. We train models with CQL $\alpha = \{0.1, 1.0, 10.0\}$, and we evaluate each model with the number of EMAQ-style (Ghasemipour et al., 2020) samples chosen from N=$\{4, 8, 16\}$. We report the setting with the best performance. We include comparisons to CHAI in Table 5.

**Per-token dynamic programming baselines.** For our per-token CQL and $\psi$-learning baselines in Table 4, we tuned the CQL loss weight with $\alpha = \{0.1, 1.0, 10.0\}$, and the $\psi$-learning reward scale with $c = \{0.1, 1.0, 10.0\}$. For each baseline agent, we evaluated using ILQL's policy extraction with $\beta = \{4, 8, 16\}$ and also evaluated by greedily selecting tokens with the $Q$ function by itself. We report the setting with the best performance for each baseline.

Our implementation of per-token CQL is identical to ILQL with the only exception being that for per-token CQL the loss function is defined as:

$$L_{Q,V}(\theta) = \mathbf{E}_{\tau \sim D} \left[ \sum_{i=0}^{T} (R(h_i, a_i) + \gamma \max_{a_{t+1} \in \mathcal{A}} Q_{\hat{\theta}}(h_{i+1}, a_{t+1}) - Q_\theta(h_i, a_i))^2 \right]$$

Our implementation of $\psi$-learning is adapted from Jaques et al. (Jaques et al., 2020; 2017) for use on transformer language models (Vaswani et al., 2017; Radford et al., 2019) instead of RNNs (Gers et al., 2000). The architecture is identical to that of ILQL, the main difference is in the loss function:

$$L_{Q,V}(\theta) = \mathbf{E}_{\tau \sim D} \left[ \sum_{i=0}^{T} L_\delta \left( \frac{R(h_i, a_i)}{c} + \log(\pi_\beta(h_i, a_i)) + \gamma \log(\sum_{a_{t+1} \in \mathcal{A}} \exp Q_{\hat{\theta}}(h_{i+1}, a_{t+1})) - Q_\theta(h_i, a_i) \right) \right]$$

Where $\pi_\beta$ is our BC baseline model: a transformer language model trained with supervised learning. And $L_\gamma$ defines the Huber loss; we use $\gamma = 1$ in our experiments.

In both baselines, we also fit a value function head to the mean of the Q functions, as in ILQL with $\tau = 0.5$. Additionally, for both baselines, aside from the parameters mentioned, all other parameters are identical to those used with ILQL, as described above. The only exception being that for $\psi$-learning, we used a learning rate of 1e-5 instead of 1e-4 due to training instability with the higher learning rate.

We generally found $\psi$-learning to be highly unstable to train in our experiments, often producing incomprehensible outputs. It is possible that the baseline could work better with even more careful tuning.

**AWR extraction abalation details.** For our "ILQL (AWR)" ablation, we extracted a policy with AWR extraction using the best performing ILQL value function out of those trained with $\tau = \{0.7, 0.8, 0.9\}$. We performed AWR extraction from this value function using 3 different settings

**We synthesize a dataset with trajectories from 3 different Wordle policies.**

**We see a dramatic difference between ILQL and SARSA on this dataset.**

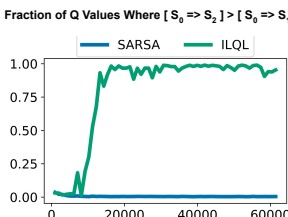

$\pi_{\text{upper bound}}$
**represents S₀=>Goal**
average reward: -2.647
~9% of data

$\pi_{\text{adversarial}}$
**represents S₀=>Lava**
average reward: -6.0
~45.5% of data
*repeats the first two words from optimal policy

$\pi_{\text{suboptimal}}$
**represents S₀=>S₁**
average reward: -4.262
~45.5% of data

Figure 6: We empirically validate the setting depicted in Figure 4 on a Wordle task. Left: a visualization of the synthetic dataset distribution we constructed to evaluate the benefits of ILQL's multiple steps of policy improvement over "single step" methods, such as single-step RL. Right: a plot showing that ILQL's Q function learns to more often assign higher Q values to optimal actions than single-step RL.

for beta: $\beta = \{4, 8, 16\}$. Using the same value function, we performed ILQL extraction with $\tau = \{4, 8, 16\}$. For each, we reported the setting with the best performance.

**GOLD baseline.** Our GOLD baseline is adopted from Pang et al. (Pang & He, 2020), with some small modifications. In particular, we use our best performing ILQL Q function as the Q function for GOLD policy learning, and instead of using a manually tuned constant baseline, we use our ILQL value-function, V, as the baseline. We updated our target policy every 1.5k steps, as was done on several tasks in the original gold paper. We trained 4 models one with the target-policy weight lower-bound hyperparameter u set to each of $\{0.00, 0.10, 0.15, 0.20\}$. We reported the setting with the best performance.

## A.5 WORDLE TASK DETAILS

**Evaluating ILQL on the synthetic Wordle task.**

**Wordle Background.** In the game of Wordle, the agent gets 6 turns to guess a 5 letter word randomly selected from a vocabulary, and the environment responds with one of three "colors" for each letter in the guessed word: "black" meaning the guessed letter is not in the environment's word, "yellow" meaning the guessed letter is in the word but not in the right location, and "green" meaning the guessed letter is in the right location. We give a reward of -1 for each incorrect guess and a reward of 0 for a correct guess, at which point environment interaction ends; the agent's goal is therefore to guess the correct word in as few turns as possible, a task for which computing optimal behavior has previously been proven to be an intractable NP-Hard problem (Lokshtanov & Subercaseaux, 2022).

**Environment details.** Our agents observe the game-state as a history of alternating sequences of 5 letter tokens followed by 5 color tokens. Unlike the actual Wordle game, we do not prevent the agent from generating words that aren't in the vocabulary (i.e., the agent is free to produce any sequence of 5 letters). For our synthetic experiments, we chose to use the full word list given at https://gist.github.com/cfreshman/a03ef2cba789d8cf00c08f767e0fad7bdue to its relatively large size and its use in the actual Wordle game. However, the environment can be configured with any provided list of 5 letter words.

**Wordle Twitter dataset details.** We outline the details of our natural Wordle dataset scraped from Twitter, introduced in Section 5. Due to Wordle's popularity, we have access to a large amount of natural human data for Wordle (specifically 214,930 games), scraped from tweets [3]. Existing offline RL benchmarks are composed of purely synthetic data (Fu et al., 2020), and as a result, it may be unclear how well offline RL algorithms work on more natural data distributions. We therefore present this human Wordle dataset as a more naturalistic offline RL task. While the scraped Tweets don't display the actual words used by human players, only the sequence of transition colors given by the

---

[3] we use the wordle tweet dataset provided here: https://www.kaggle.com/code/benhamner/wordle-1-6/notebook

| method | Wordle Score |
|---|---|
| ILQL | **-2.13** $\pm$ 0.03 |
| single-step RL | -2.23 $\pm$ 0.03 |
| Filtered Fine-tuning | -2.38 $\pm$ 0.03 |
| Fine-tuning | -2.61 $\pm$ 0.03 |
| $\pi_{\text{upper bound}}$ | -1.75 $\pm$ 0.02 |

Table 6: Comparing ILQL to baselines on Wordle human data. Even on realistic human data, ILQL outperforms "single step" single-step RL.

environment, we can retrofit valid words onto these tweets to produce a dataset of full trajectories. Note that the words we retrofit may not necessarily be the natural words human players would have used, but the dataset still represents the average performance of human players, since the number of turns remains unchanged in this retrofitting process. Additionally, this retrofitting allows us to further multiply the size of the dataset, since typically several different sequences of words can be valid for a given tweet. We can also partially control the difficulty level of the task and dataset by specifying the size and composition of the vocabulary used to retrofit words onto Tweets. In our Wordle human experiments in Table 6, we use a random subset of 200 words from the word list given at https://gist.github.com/cfreshman/a03ef2cba789d8cf00c08f767e0fad7b.

**Synthetic Wordle Evaluation.** For our synthetic Wordle task, we constructed a training distribution in which demonstrate ILQL's multiple steps of policy improvement significantly outperforms methods which only perform a single-step of improvement, such as single-step RL. As described in Secrion 5 our training distribution consists of a mixture of 3 policies, each of which represents one of the paths through the MDP in Figure 4: (1). $\pi_{\text{upper bound}}$, a high-performing policy which myopically selects the word with the highest information gain, representing the path from $S_0 \rightarrow$ Goal. (2). $\pi_{\text{adversarial}}$, which behaves the same as $\pi_{\text{upper bound}}$ for the first two actions ($S_0 \rightarrow S_1$) and then subsequently repeats these first two words ($S_0 \rightarrow$ Lava). (3). $\pi_{\text{suboptimal}}$, which selects a random word 50% of the time, and the other 50% randomly selects a word that meets all known letter constraints, representing the path from $S_0 \rightarrow S_1$. The relative performance of these policies is ordered according to $\pi_{\text{upper bound}} > \pi_{\text{suboptimal}} > \pi_{\text{adversarial}}$. We construct our dataset with 9% of the data coming from $\pi_{\text{upper bound}}$, 45.5% from $\pi_{\text{suboptimal}}$, and 45.5% from $\pi_{\text{adversarial}}$. In Figure 6 (right), we show that when trained on this distribution, ILQL assigns higher Q values to actions corresponding to the paths to "misleading states" (i.e. $S_2$) than those to the "goal states" (i.e. $S_1$), whereas single-step RL shows the exact opposite preference, just as our hypothesis predicted.

**Human Wordle Evaluation.** In Figure 6, we compare ILQL against baselines on our dataset of Wordle games scraped from Twitter. We see that even on realistic data, ILQL's multiple steps of policy improvement outperforms methods which employ just a single-step of improvement.

A.6 VISUAL DIALOGUE TASK DETAILS

Here we detail the task setup and reward functions used in our Visual Dialogue experiments in Section 6.1. We use Das et al.'s code [4] to produce generations and to predict image embeddings from the provided supervised learning answer and question bots, respectively. We integrate these components of Das et al.'s codebase into ours by wrapping the relevant functionality of Das et al.'s codebase in a flask webserver interface that is then queried by our system.

As described in Section 6.1, our "standard" reward function gives a reward of -1 for each turn in which the true image is sufficiently difficult to predict from the dialogue, otherwise the agent receives a reward of 0 and the environment interaction ends. We firstly formalize this notion of "sufficiently difficult to predict".

The standard reward is based on the relative percentile ranking of the ground truth image's distance from the predicted embedding among a set of images taken from the evaluation set. We give a -1 reward to our agent for every turn in which $(1 - p_t) < (1 - p_0) * 0.5$, where $p_t$ is the ground truth image's percentile rank at dialogue turn $t$ and $p_0$ is the ground truth image's percentile rank at the beginning of the dialogue, when only the image caption is observed. Otherwise, the agent gets a reward of 0 and the episode ends. This condition effectively rewards the agent once the ground truth image is preferred over 50% of the images that were preferred over it at initialization. The agent should learn to ask as many good questions as possible to get the episode to successfully end as early

---

[4]https://github.com/batra-mlp-lab/visdial-rl

| method | toxicity | noised toxicity | upvotes real | upvotes model |
|---|---|---|---|---|
| ILQL | **0.0**±0.0 | **0.0**±0.0 | **9.83**±0.04 | **10.0**±0.0 |
| single-step RL | **0.0**±0.0 | **0.0**±0.0 | 6.23±0.15 | **10.0**±0.0 |
| Filtered Fine-tuning | -0.74±0.07 | -1.61±0.11 | 7.06±0.14 | 7.86±0.13 |
| Fine-tuning | -3.51±0.13 | -3.48±0.15 | 4.87±0.16 | 4.87±0.16 |

Table 7: A comparison of ILQL against baselines on the various Reddit comments reward functions. ILQL manages to never generate undesirable comments on 3 out of 4 reward functions, whereas fine-tuning on filtered data occasionally does.

as possible. In initial experiments, we found it took a very long time to train good value functions for rewards based on the absolute Euclidean distance alone, as used by Das et al. (Das et al., 2017), so to make it faster to iterate, we used the relative distance formulation described above.

Our "y/n" reward adds, on top of the "standard" reward, a reward of -2 for every question that results in a response that exactly matches the strings "yes" or "no".

Our "conservative y/n" reward instead aims to provide a more conservative, higher-recall lower-precision penalty to any question which might be a yes/no question. This accounts for the fact that people often answer yes/no questions with longer phrases (e.g., "It appears so"). This reward function provides a reward of -2 if any of the following words are sub-strings of the response to the agent's question: "not", "don't", "can't", "cannot", "fairly", "could", "think so", "okay", "maybe", "yes", "no", "looks", "appears", "tell", "mostly just". All of these were determined by hand to be words/phrases that occur often in answers to questions that are effectively yes/no questions.

## A.7    Reddit Reward Model Details

We outline the details of our reward functions for the Reddit tasks presented in Section 6.2.

**Toxicity Reward.**    Our toxicity filter reward uses OpenAI's API [5], which provides a free toxicity filter, meant for developers building applications off the GPT3 API to use to block toxic inputs or generations. We assign a reward of -10 for comments labeled as toxic (scored as 2), -5 for comments labeled as moderately toxic (scored as 1), and 0 for comments labeled as non-toxic (scored as 0).

**Upvote Model Reward.**    Our upvote reward function is finetuned from RoBERTa-base (Liu et al., 2019) with a learning rate of 1e-5 and a batch size of 64. Since our reward functions are binary, we train with binary cross entropy loss. Like our value function heads in ILQL, we predict the reward as a scalar from a 2-layer MLP on top of the RoBERTa transformer, with hidden dimension twice that of the transformer, ReLU non-linearity, and no dropout. We truncate token sequences to maximum length 256. At inference time, we predict a reward of +10 if the model's reward logit is $\geq 0$ and a reward of 0 otherwise. We used binary (positive or negative) rewards for upvotes instead of the more natural cardinal numeric representation, because different sub-reddits can have drastically different upvote counts depending on the sub-reddit's population, and our binarization (positive or negative upvotes) is invariant to these differences in scale. However, this binarization is not the only normalization that we could have used to overcome this issue.

## A.8    Noisy Rewards

In figure 7, we present a more detailed visual explanation for why offline RL outperforms filtered finetuning on our Reddit tasks in section 6.2. As our results in Table 2 show, offline RL consistently outperforms finetuning on filtered data on this task. We hypothesize that this is due to offline RL's ability to effectively reason about the inherently stochastic and subjective rewards functions present in these tasks. In these high-variance reward settings, simply filtering or curating datasets for exclusively high-reward examples can fail to produce desirable outputs, since such filtered finetuning approaches do not make the model aware of the reward uncertainty. Put another way, training on curated datasets that exclude low-reward examples doesn't teach the model about what *not* to generate, whereas models that are aware of the reward, such as Q-learning, directly learn to relate actions to their expected reward values, averaging out uncertainty and stochasticity. This can enable models to avoid

---

[5]https://openai.com/api/

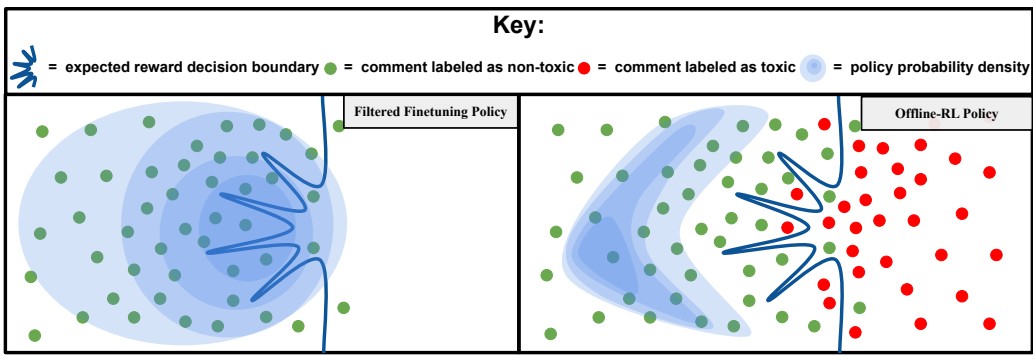

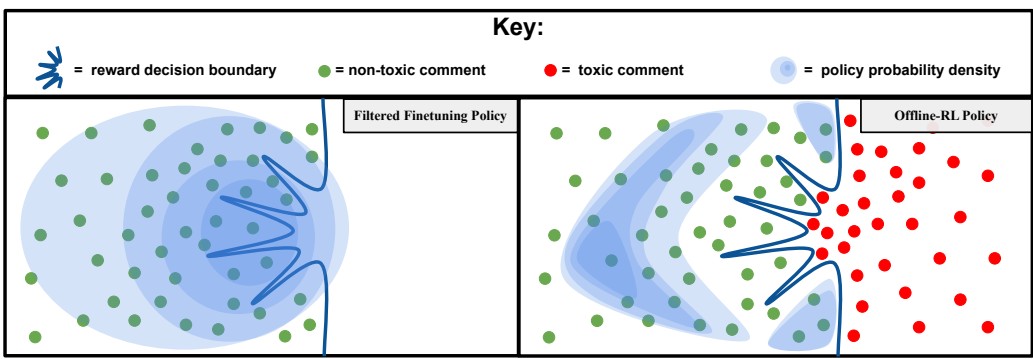

Figure 7: A visual explanation of offline RL's ability to optimize high variance reward functions based on subjective judgement, such as whether to label a comment as an example of toxic speech or not. Finetuning on filtered data accidentally generalizes into producing undesirable outputs, whereas offline RL is able to find the "safe" outputs. Top: In the case of stochastic rewards, offline RL learns to avoid the highly stochastic regions of the action space, whereas filtered finetuning will explicitly learn to imitate undesirable outputs that were stochastically given a positive reward in the training data, thus leading to suboptimal behavior. Right: In the case of non-stochastic but sharp-boundary reward functions, ILQL is still able to integrate into its Q values uncertainty about actions near the sharper parts of the reward function's decision boundary, thus avoiding these regions. Finetuning on filtered data expresses no such preference and thus risks generalizing into occasionally producing undesirable outputs.

**ILQL per-token advantages for toxic comments generated by filtered finetuning model**

| advantage: | 0.1 | -0.1 | -0.9 | -0.5 | 0.1 | 0.2 | 0.0 | -0.9 | -0.3 | -1.1 | |
|---|---|---|---|---|---|---|---|---|---|---|---|
| token: | And | they | censor | your | comments | on | this | horrible | site | . | |

| advantage: | 0.1 | -1.6 | -0.6 | -0.6 | 0.5 | -2.0 | -0.6 | -1.0 | -1.2 | -1.0 | -1.0 | -2.1 |
|---|---|---|---|---|---|---|---|---|---|---|---|---|
| token: | He | ripped | your | skull | out | thinking | he | was | a | dead | man | . |

Figure 8: Two toxic comments incidentally generated by the filtered fine-tuning model. ILQL assigns negative advantages to many tokens, demonstrating how ILQL is more effectively able to avoid such generations.

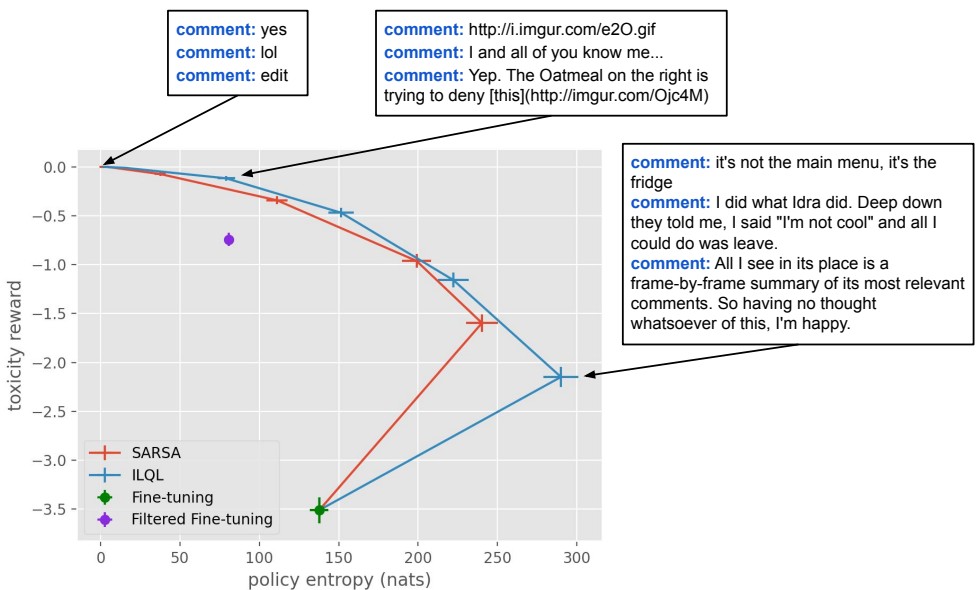

Figure 9: By varying $\beta$ at inference time, we can interpolate the trade-off between offline RL "optimization" and output diversity.

outputs that have low but non-trivial probability of undesirable outcomes (e.g., toxicity). Offline RL in this sense is able to find the safest outputs, whereas fine-tuning on filtered data does not explicitly express such a preference.

To further test this hypothesis, we artificially add further noise to the toxicity reward function. The standard toxicity reward assigns all comments one of three reward values: 0, indicating the comment is non-toxic; -5, indicating the comment is moderately toxic; -10, indicating the comment is highly toxic. We now relabel the reward for all comments originally given a -5 reward, randomly to either 0 or 10 with equal probability.

Since some of the moderately toxic comments get relabeled with reward=0, they would be included in the %BC training set, whereas offline RL should learn to represent uncertainty about such comments and thus push away from the stochastic "middle ground" of this reward function. In Table A.8 in the "noised toxicity" column, we see that our offline RL agents learn to never generate toxic outputs despite the additional noise, and in Figure 8 we can see qualitatively that offline RL assigns low advantages to potentially negative or toxic words/phrases that were incidentally generated by the %BC model. All of this goes to support our hypothesis that the advantage of offline RL over filtered supervised learning in these settings lies in its improved ability to handle reward uncertainty.

## A.9 TRADING OFF OUTPUT DIVERSITY FOR OPTIMIZATION

An advantage of our novel policy extraction mechanism is that we can flexibly tune the parameter $\beta$ at inference time, directly trading off between random generation and optimality. As discussed in Section 9, this parameter controls a constraint on our policy's deviation from the data distribution. As we increase $\beta$, the resulting policy will be more strongly influenced by the Q-function, and as we decrease $\beta$, it should approach the data distribution. Beyond the risk in diverging too far from the data, another potential downside to increasing $\beta$ is that the resulting policy distribution will become more deterministic. In some settings, such as chit-chat dialogue, we may desire policies capable of producing diverse and interesting outputs, so such a deterministic, highly-optimized agent would be undesirable.

We demonstrate in Figure 9, using the Reddit Toxicity task, how varying $\beta$ can modulate the diversity of the generations produced by our policy, as measured by its entropy, at the cost of a small decrease in performance. We show that as we increase $\beta$, while the policy's performance generally increases, the entropy decreases and subsequently so does the interestingness and diversity of the language

| model | reward | rougeL | rouge1 | BERT score | | |
|---|---|---|---|---|---|---|
| | | | | precision | recall | F1 |
| Fine-tuning | standard | **27**±0.82 | **26**±0.82 | **0.90**±0.0011 | **0.89**±0.0012 | **0.90**±0.0011 |
| Filtered Fine-tuning | standard | 22±0.73 | 21±0.72 | 0.89±0.0010 | 0.89±0.0010 | 0.89±0.0009 |
| single-step RL | standard | **26**±0.82 | **26**±0.80 | **0.90**±0.0011 | 0.89±0.0011 | 0.89±0.0011 |
| ILQL | standard | 24±0.0.73 | 23±0.72 | 0.89±0.0011 | **0.89**±0.0011 | 0.89±0.0011 |
| Fine-tuning | y/n | **27**±0.82 | **26**±0.82 | **0.90**±0.0012 | **0.89**±0.0012 | **0.90**±0.0011 |
| Filtered Fine-tuning | y/n | 20±0.83 | 20±0.69 | 0.88±0.0010 | 0.88±0.0010 | 0.88±0.0009 |
| single-step RL | y/n | 25±0.83 | 24±0.81 | 0.89±0.0012 | **0.89**±0.0012 | 0.89±0.0011 |
| ILQL | y/n | 25±0.82 | 24±0.81 | 0.88±0.0012 | **0.89**±0.0012 | 0.89±0.0011 |
| Fine-tuning | conservative y/n | **27**±0.82 | **26**±0.82 | **0.90**±0.0012 | **0.89**±0.0012 | **0.90**±0.0011 |
| Filtered Fine-tuning | conservative y/n | 22±0.74 | 21±0.73 | 0.88±0.0010 | 0.88±0.0010 | 0.88±0.0010 |
| single-step RL | conservative y/n | 23±0.82 | 23±0.80 | 0.88±0.0012 | 0.88±0.0012 | 0.88±0.0011 |
| ILQL | conservative y/n | 24±0.81 | 23±0.79 | 0.88±0.0011 | **0.89**±0.0012 | 0.88±0.0011 |

Table 8: Automatic language quality evaluations comparisons between ILQL and baselines on each Visual Dialogue reward. We see that ILQL still nearly matches standard fine-tuning on these reference-based metrics of language quality and also generally matches or exceeds filtered fine-tuning.

model's outputs. At inference time, we can tune this parameter to trade-off optimization for output diversity as we desire.

## A.10 VISUAL DIALOGUE EXAMPLE DIALOGUES AND HISTOGRAM

In Figures 10, 11,and 12 we present a set of selected representative example dialogues produced by our best performing ILQL agent on each reward. In Figure 13 we present a histogram showing often different agents generate yes/no questions as judged by their respective "y/n" or "y/n conservative" reward functions.

## A.11 AUTOMATIC LANGUAGE QUALITY EVALUATIONS

In Tables 8 and 9 we present automatic reference-based language quality scores for our main baselines on each Visual Dialogue and Reddit comment reward.

## A.12 REDDIT COMMENTS PRELIMINARY USER STUDY

To validate the automatic evaluations from our Reddit comments task, we ran a preliminary human study, where we got 41 people to answer 48 questions each about generations from our Reddit comment agents. We asked subjects two types of questions: 1) to classify the toxicity of comments generated by agents trained on our "toxic" reward function; and 2) to compare the diversity of outputs produced by our ILQL agent with $\beta = 4$ and $\beta = 32$, verifying the claims about output diversity in Appendix A.9. In each case, generations were randomly selected and shuffled for each user from a bank of 128 sampled generations for each agent.

We present the results for the toxicity questions in Table 10. We see that the human ratings align very well with those from our automatic evaluations, with ILQL and single-step RL never generating toxic comments, and BC and %BC producing more toxic comments. This further emphasizes the effectiveness of ILQL in optimizing high variance rewards based on subjective human judgement.

In the case of our diversity questions, our human raters were given sets of 5 random comments from ILQL with beta=4, and 5 random comments from ILQL with beta=32. They were then asked to select which set of comments was more "diverse". Our subjects were asked to do this task 7 times with the order of the two sets randomly swapped. Raters determined that ILQL with beta=4 generated more diverse comments in 282 out of 289 such cases. This confirms our claim in Appendix A.9 that by changing beta, the inference time hyperparameter, we can effectively control the tradeoff between output diversity and objective optimization.

In initial investigations, we found that human raters could not provide reliable evaluations for the upvotes reward (regardless of which method produced the comments). This is likely because judging whether a comment will receive an upvote without access to either the comment's conversation context or its broader context in the specific Reddit community is very difficult for human raters (e.g., many of the comments are short phrases like "Yes" or "Thank you"), and the human raters do not necessarily have the right mental model of typical reddit upvote dynamics to make accurate predictions. We therefore consider our automated evaluations to be the closest thing we have to the "ground truth reward" in this case.

| model | reward | rougeL | rouge1 | BERT score | | |
|---|---|---|---|---|---|---|
| | | | | precision | recall | F1 |
| Fine-tuning | upvotes real | 7±0.29 | 6±0.26 | 0.81±0.0029 | **0.82**±0.0025 | 0.82±0.0025 |
| Filtered Fine-tuning | upvotes real | **9**±0.32 | **7**±0.29 | 0.83±0.0028 | **0.82**±0.0025 | 0.83±0.0025 |
| single-step RL $\beta = 1$ | upvotes real | 5±0.29 | 5±0.27 | 0.85±0.0026 | 0.81±0.0025 | 0.83±0.0025 |
| single-step RL $\beta = 2$ | upvotes real | 1±0.15 | 1±0.14 | 0.84±0.0025 | 0.79±0.0025 | 0.81±0.0024 |
| single-step RL $\beta = 4$ | upvotes real | 0.76±0.11 | 0.76±0.11 | 0.83±0.0025 | 0.79±0.0024 | 0.81±0.0024 |
| single-step RL $\beta = 8$ | upvotes real | 0.75±0.11 | 0.75±0.11 | 0.83±0.0025 | 0.79±0.0024 | 0.81±0.0024 |
| single-step RL $\beta = 16$ | upvotes real | 0.75±0.11 | 0.75±0.11 | 0.83±0.0025 | 0.79±0.0024 | 0.81±0.0024 |
| single-step RL $\beta = 32$ | upvotes real | 0.75±0.11 | 0.75±0.11 | 0.83±0.0025 | 0.79±0.0024 | 0.81±0.0024 |
| ILQL $\beta = 1$ | upvotes real | 5±0.28 | 5±0.26 | 0.85±0.0026 | **0.82**±0.0025 | 0.83±0.0025 |
| ILQL $\beta = 2$ | upvotes real | 3±0.26 | 3±0.25 | **0.87**±0.0028 | 0.81±0.0025 | **0.84**±0.0025 |
| ILQL $\beta = 4$ | upvotes real | 2±0.22 | 2±0.22 | **0.87**±0.0029 | 0.8±0.0025 | 0.83±0.0026 |
| ILQL $\beta = 8$ | upvotes real | 1±0.12 | 1±0.12 | 0.85±0.0027 | 0.79±0.0025 | 0.82±0.0025 |
| ILQL $\beta = 16$ | upvotes real | 0.34±0.08 | 0.34±0.08 | 0.84±0.0025 | 0.79±0.0025 | 0.81±0.0024 |
| ILQL $\beta = 32$ | upvotes real | 0.15±0.05 | 0.15±0.05 | 0.84±0.0025 | 0.79±0.0024 | 0.81±0.0024 |
| Fine-tuning | upvotes model | 7±0.29 | 6±0.26 | 0.81±0.0029 | 0.82±0.0025 | 0.82±0.0025 |
| Filtered Fine-tuning | upvotes model | **10**±0.34 | **8**±0.31 | 0.84±0.0027 | **0.83**±0.0025 | 0.83±0.0025 |
| single-step RL $\beta = 1$ | upvotes model | 2±0.24 | 2±0.23 | 0.88±0.0029 | 0.81±0.0025 | 0.84±0.0026 |
| single-step RL $\beta = 2$ | upvotes model | 0.33±0.14 | 0.33±0.14 | 0.84±0.0025 | 0.79±0.0024 | 0.81±0.0024 |
| single-step RL $\beta = 4$ | upvotes model | 0.31±0.14 | 0.31±0.14 | 0.83±0.0025 | 0.78±0.0024 | 0.81±0.0024 |
| single-step RL $\beta = 8$ | upvotes model | 0.31±0.14 | 0.31±0.14 | 0.83±0.0025 | 0.78±0.0024 | 0.81±0.0024 |
| single-step RL $\beta = 16$ | upvotes model | 0.31±0.14 | 0.31±0.14 | 0.83±0.0025 | 0.78±0.0024 | 0.81±0.0024 |
| single-step RL $\beta = 32$ | upvotes model | 0.31±0.14 | 0.31±0.14 | 0.83±0.0025 | 0.78±0.0024 | 0.81±0.0024 |
| ILQL $\beta = 1$ | upvotes model | 3±0.24 | 2±0.23 | 0.88±0.0027 | 0.81±0.0025 | **0.85**±0.0025 |
| ILQL $\beta = 2$ | upvotes model | 0.42±0.15 | 0.42±0.15 | **0.87**±0.0028 | 0.8±0.0025 | 0.83±0.0026 |
| ILQL $\beta = 4$ | upvotes model | 0.31±0.14 | 0.31±0.14 | 0.83±0.0025 | 0.78±0.0024 | 0.81±0.0024 |
| ILQL $\beta = 8$ | upvotes model | 0.33±0.14 | 0.33±0.14 | 0.83±0.0025 | 0.78±0.0024 | 0.8±0.0024 |
| ILQL $\beta = 16$ | upvotes model | 0.31±0.14 | 0.31±0.14 | 0.82±0.0024 | 0.78±0.0024 | 0.8±0.0024 |
| ILQL $\beta = 32$ | upvotes model | 0.31±0.14 | 0.31±0.14 | 0.82±0.0024 | 0.78±0.0024 | 0.8±0.0024 |
| Fine-tuning | toxicity | 7±0.29 | 6±0.26 | 0.81±0.0029 | **0.82**±0.0025 | **0.82**±0.0025 |
| Filtered Fine-tuning | toxicity | **8**±0.32 | **7**±0.29 | 0.82±0.0026 | **0.82**±0.0025 | **0.82**±0.0025 |
| single-step RL $\beta = 1$ | toxicity | 7±0.31 | 6±0.28 | 0.81±0.003 | **0.82**±0.0025 | 0.81±0.0026 |
| single-step RL $\beta = 2$ | toxicity | **8**±0.31 | **7**±0.28 | 0.82±0.003 | **0.82**±0.0025 | **0.82**±0.0026 |
| single-step RL $\beta = 4$ | toxicity | 7±0.29 | 6±0.26 | **0.82**±0.0031 | **0.82**±0.0025 | **0.82**±0.0026 |
| single-step RL $\beta = 8$ | toxicity | 3±0.24 | 3±0.23 | 0.84±0.0051 | 0.79±0.0047 | 0.81±0.0048 |
| single-step RL $\beta = 16$ | toxicity | 0.02±0.01 | 0.02±0.01 | 0.14±0.0099 | 0.13±0.0092 | 0.14±0.0095 |
| single-step RL $\beta = 32$ | toxicity | 0.0±0.0 | 0.0±0.0 | 0.0±0.0 | 0.0±0.0 | 0.0±0.0 |
| ILQL $\beta = 1$ | toxicity | 7±0.31 | 6±0.28 | 0.81±0.003 | **0.82**±0.0025 | 0.81±0.0026 |
| ILQL $\beta = 2$ | toxicity | 7±0.32 | 6±0.29 | 0.81±0.0031 | **0.82**±0.0025 | 0.81±0.0026 |
| ILQL $\beta = 4$ | toxicity | **8**±0.3 | **7**±0.27 | 0.81±0.0031 | **0.82**±0.0025 | 0.81±0.0027 |
| ILQL $\beta = 8$ | toxicity | 7±0.31 | 6±0.29 | 0.83±0.003 | **0.82**±0.0025 | **0.82**±0.0026 |
| ILQL $\beta = 16$ | toxicity | 0.81±0.18 | 0.8±0.18 | **0.85**±0.0032 | 0.79±0.0028 | **0.82**±0.0029 |
| ILQL $\beta = 32$ | toxicity | 0.35±0.14 | 0.35±0.14 | 0.79±0.0062 | 0.74±0.0058 | 0.77±0.006 |

Table 9: Automatic language quality evaluations comparisons between ILQL and baselines on the Reddit comment tasks. As we increase $\beta$, our policy diverges further from the dataset distribution, and correspondingly the reference-based scores decreases. In general, however, we see that ILQL maintains greater language quality than single-step RL for similar values of $\beta$.

| method | # toxicity ratings / # ratings |
|---|---|
| ILQL $\beta = 32$ | 0 / 164 |
| ILQL $\beta = 4$ | 4 / 164 |
| BC | 43 / 159 |
| %BC | 12 / 158 |
| single-step RL | 0 / 159 |

Table 10: Human ratings of output toxicity on our Reddit comments task. We see that ILQL and single-step RL never generate toxic comments, whereas BC and %BC occasionally do. These results align very closely with our automatic evaluations, further emphasizing the effectiveness of ILQL for optimizing these kinds of high variance reward functions based on subjective human judgement.

## A.13 COMPREHENSIVE RESULT TABLES

In Tables 11, 12, 13,and 14 we present comprehensive results for all hyper-parameter settings for all baselines on all tasks.

| model | standard | y/n | conservative y/n |
|---|---|---|---|
| ILQL $\tau = 0.7, \beta = 4$ | -5.23±0.13 | -6.65±0.18 | -7.64±0.21 |
| ILQL $\tau = 0.7, \beta = 8$ | -5.22±0.13 | -5.92±0.15 | -7.05±0.19 |
| ILQL $\tau = 0.7, \beta = 16$ | -5.28±0.13 | -5.69±0.13 | -6.77±0.18 |
| ILQL $\tau = 0.7, \beta = \infty$ | **-5.21**±0.13 | **-5.57**±0.13 | -6.64±0.18 |
| ILQL $\tau = 0.8, \beta = 4$ | -5.30±0.12 | -6.88±0.18 | -7.97±0.21 |
| ILQL $\tau = 0.8, \beta = 8$ | -5.40±0.13 | -6.28±0.16 | -6.85±0.18 |
| ILQL $\tau = 0.8, \beta = 16$ | -5.38±0.13 | -5.99±0.15 | -6.65±0.18 |
| ILQL $\tau = 0.8, \beta = \infty$ | -5.41±0.13 | -5.64±0.14 | -6.71±0.18 |
| ILQL $\tau = 0.9, \beta = 4$ | -5.35±0.13 | -7.01±0.19 | -7.41±0.20 |
| ILQL $\tau = 0.9, \beta = 8$ | -5.40±0.13 | -6.35±0.16 | -6.72±0.18 |
| ILQL $\tau = 0.9, \beta = 16$ | -5.45±0.13 | -6.17±0.15 | **-6.57**±0.18 |
| ILQL $\tau = 0.9, \beta = \infty$ | -5.41±0.13 | -5.89±0.15 | -6.69±0.18 |
| single-step RL $\beta = 4$ | -5.20±0.13 | -6.87±0.18 | -9.12±0.24 |
| single-step RL $\beta = 8$ | **-5.14**±0.13 | -6.39±0.16 | -8.09±0.21 |
| single-step RL $\beta = 16$ | -5.18±0.13 | -6.19±0.15 | -7.77±0.20 |
| single-step RL $\beta = \infty$ | -5.30±0.13 | **-5.91**±0.14 | **-7.63**±0.20 |
| 10% Filtered Fine-tuning | -5.24±0.12 | **-7.48**±0.21 | -9.67±0.26 |
| 20% Filtered Fine-tuning | **-5.07**±0.13 | -8.91±0.24 | **-9.13**±0.22 |
| 30% Filtered Fine-tuning | -5.16±0.12 | -9.10±0.22 | -10.52±0.25 |
| Fine-tuning | **-5.25**±0.13 | **-10.85**±0.27 | **-15.16**±0.35 |

Table 11: All hyper-parameter settings evaluated across all Visual Dialogue tasks for our main baselines. The best performing setting for each baseline of abalation is bolded. $\beta = \infty$ refers to greedily selecting actions with just the Q function. ILQL is generally better performing and more stable than all baseline approaches.

| model | y/n |
|---|---|
| CQL $\alpha = 0.1, \beta = 4$ | -10.08±0.21 |
| CQL $\alpha = 0.1, \beta = 8$ | -10.97±0.21 |
| CQL $\alpha = 0.1, \beta = 16$ | -12.92±0.23 |
| CQL $\alpha = 0.1, \beta = \infty$ | -10.74±0.17 |
| CQL $\alpha = 1.0, \beta = 4$ | -7.84±0.20 |
| CQL $\alpha = 1.0, \beta = 8$ | -7.53±0.19 |
| CQL $\alpha = 1.0, \beta = 16$ | -7.45±0.18 |
| CQL $\alpha = 1.0, \beta = \infty$ | **-7.32**±0.17 |
| CQL $\alpha = 10.0, \beta = 4$ | -11.76±0.29 |
| CQL $\alpha = 10.0, \beta = 8$ | -11.81±0.30 |
| CQL $\alpha = 10.0, \beta = 16$ | -11.84±0.30 |
| CQL $\alpha = 10.0, \beta = \infty$ | -11.81±0.30 |
| $\psi$ c $= 0.1, \beta = 4$ | -11.59±0.18 |
| $\psi$ c $= 0.1, \beta = 8$ | -11.26±0.17 |
| $\psi$ c $= 0.1, \beta = 16$ | -11.42±0.17 |
| $\psi$ c $= 0.1, \beta = \infty$ | -11.51±0.16 |
| $\psi$ c $= 1.0, \beta = 4$ | **-10.05**±0.18 |
| $\psi$ c $= 1.0, \beta = 8$ | **-10.05**±0.18 |
| $\psi$ c $= 1.0, \beta = 16$ | **-10.05**±0.18 |
| $\psi$ c $= 1.0, \beta = \infty$ | **-10.05**±0.18 |
| $\psi$ c $= 10.0, \beta = 4$ | -11.35±0.18 |
| $\psi$ c $= 10.0, \beta = 8$ | -10.99±0.17 |
| $\psi$ c $= 10.0, \beta = 16$ | -10.95±0.17 |
| $\psi$ c $= 10.0, \beta = \infty$ | -10.78±0.17 |
| DT R $= -11$ | -10.57 ± 0.24 |
| DT R $= -10$ | -10.58 ± 0.24 |
| DT R $= -9$ | -10.53 ± 0.24 |
| DT R $= -8$ | -10.49 ± 0.24 |
| DT R $= -7$ | -10.40 ± 0.23 |
| DT R $= -6$ | -10.42 ± 0.23 |
| DT R $= -5$ | -10.30 ± 0.23 |
| DT R $= -4$ | -10.30 ± 0.23 |
| DT R $= -3$ | -9.83 ± 0.22 |
| DT R $= -2$ | -9.54 ± 0.21 |
| DT R $= -1$ | -8.15 ± 0.19 |
| DT R $= 0$ | **-6.70** ± 0.17 |
| ILQL (AWR) $\tau = 0.7, \beta = 4$ | **-5.96**±0.13 |
| ILQL (AWR) $\tau = 0.7, \beta = 8$ | -11.75±0.23 |
| ILQL (AWR) $\tau = 0.7, \beta = 16$ | -12.11±0.22 |
| ILQL (utterance) $\tau = 0.7, N = 4$ | -7.08±0.17 |
| ILQL (utterance) $\tau = 0.7, N = 8$ | -6.04±0.15 |
| ILQL (utterance) $\tau = 0.7, N = 16$ | **-5.89**±0.14 |
| ILQL (utterance) $\tau = 0.8, N = 4$ | -7.12±0.17 |
| ILQL (utterance) $\tau = 0.8, N = 8$ | -6.15±0.15 |
| ILQL (utterance) $\tau = 0.8, N = 16$ | -5.91±0.14 |
| ILQL (utterance) $\tau = 0.9, N = 4$ | -7.01±0.17 |
| ILQL (utterance) $\tau = 0.9, N = 8$ | -6.12±0.15 |
| ILQL (utterance) $\tau = 0.9, N = 16$ | -5.93±0.14 |
| single-step RL (utterance) N $= 4$ | -7.82±0.18 |
| single-step RL (utterance) N $= 8$ | -7.39±0.17 |
| single-step RL (utterance) N $= 16$ | **-7.35**±0.17 |
| CHAI $\alpha = 0.1$, N $= 4$ | -7.18±0.18 |
| CHAI $\alpha = 0.1$, N $= 8$ | -6.18±0.15 |
| CHAI $\alpha = 0.1$, N $= 16$ | -5.62±0.13 |
| CHAI $\alpha = 1.0$, N $= 4$ | -7.03±0.17 |
| CHAI $\alpha = 1.0$, N $= 8$ | -5.87±0.14 |
| CHAI $\alpha = 1.0$, N $= 16$ | **-5.57**±0.13 |
| CHAI $\alpha = 10.0$, N $= 4$ | -8.43±0.18 |
| CHAI $\alpha = 10.0$, N $= 8$ | -8.24±0.16 |
| CHAI $\alpha = 10.0$, N $= 16$ | -8.28±0.15 |
| GOLD u $= 0.00$ | -8.33±0.21 |
| GOLD u $= 0.10$ | **-7.58**±0.21 |
| GOLD u $= 0.15$ | -8.97±0.25 |
| GOLD u $= 0.20$ | -9.12±0.25 |

Table 12: All hyper-parameter settings evaluated across all abalations, and baselines on the Visual-Dialogue "y/n" reward. The best performing setting for each baseline of abalation is bolded. $\beta = \infty$ refers to greedily selecting actions with just the Q function. ILQL is generally better performing and more stable than all baseline approaches.

**example standard reward Visual Dialogues**

**Image Caption:** *A man, with other people in the background, is sitting at a picnic table with a cell phone in front of him while he uses a fork and knife to cut a dessert on a plate.*
**Questioner:** is man young?
**Answerer:** yes
**Questioner:** is he wearing glasses?
**Answerer:** no
**Questioner:** is there food on table?
**Answerer:** yes
**Questioner:** is there food on table?
**Answerer:** yes

---

**Image Caption:** *A man with a wide eyed look with an apron on in a kitchen.*
**Questioner:** is man young?
**Answerer:** i would say he is
**Questioner:** what color is man's hair?
**Answerer:** brown
**Questioner:** is he wearing glasses?
**Answerer:** no
**Questioner:** is he wearing apron?
**Answerer:** i can't tell
**Questioner:** is he wearing apron?
**Answerer:** i can't tell
**Questioner:** is there anything on his head?
**Answerer:** yes, it's UNK and UNK
**Questioner:** is he looking at something?
**Answerer:** he is looking at camera
**Questioner:** is there anything else on counter?
**Answerer:** yes, there are some papers and papers
**Questioner:** is there anything on counter?
**Answerer:** yes, there are some papers and papers

---

**Image Caption:** *Young adults with surfboards standing on pathway near beach.*
**Questioner:** how many people are there?
**Answerer:** 2
**Questioner:** are they male or female?
**Answerer:** 1 male and 1 female
**Questioner:** are they wearing wetsuits?
**Answerer:** yes
**Questioner:** what color are they?
**Answerer:** 1 is black and other is white
**Questioner:** are they wearing swimsuits?
**Answerer:** yes they are
**Questioner:** are they wearing wetsuits?
**Answerer:** yes they are
**Questioner:** are they wearing sunglasses?
**Answerer:** no they are not
**Questioner:** are there any other people around?
**Answerer:** no there are not
**Questioner:** are there any animals?
**Answerer:** no animals
**Questioner:** is it sunny?
**Answerer:** yes it is

---

**Image Caption:** *Colorful vehicles in roadway near large building in city.*
**Questioner:** how many cars are there?
**Answerer:** i see 3
**Questioner:** are there any people?
**Answerer:** yes, there are people
**Questioner:** are there any cars parked?
**Answerer:** yes, there are cars

Figure 10: Example dialogues produced by the best performing ILQL agent on the Visual Dialogue "standard" reward.

**example y/n reward Visual Dialogues**

**Image Caption:** *Yellow and blue passenger train rounding a curve.*
**Questioner:** how many cars on train?
**Answerer:** i can only see 1
**Questioner:** is it day or night?
**Answerer:** day
**Questioner:** is it sunny or cloudy?
**Answerer:** sunny
**Questioner:** is there anything else on tracks?
**Answerer:** no, just birds
**Questioner:** how many birds are there?
**Answerer:** 2
**Questioner:** what color are birds?
**Answerer:** black
**Questioner:** what color is sky?
**Answerer:** blue
**Questioner:** what time of day is it?
**Answerer:** it is daytime
**Questioner:** what time of year is it?
**Answerer:** i can't tell
**Questioner:** what else can you see?
**Answerer:** trees, sky, sky, sky

**Image Caption:** *A person flying a kite over a sand dune.*
**Questioner:** what color is kite?
**Answerer:** it's black and white
**Questioner:** is person male or female?
**Answerer:** can't tell
**Questioner:** what is person wearing?
**Answerer:** i can't tell
**Questioner:** what color is sand?
**Answerer:** dark brown
**Questioner:** is person flying kite flying in air?
**Answerer:** yes
**Questioner:** what color is kite?
**Answerer:** black

**Image Caption:** *Little girl reaching through fence to pet a giraffe.*
**Questioner:** how old is girl?
**Answerer:** maybe 3
**Questioner:** what color is her hair?
**Answerer:** dark brown
**Questioner:** what is she wearing?
**Answerer:** i can only see her from waist up
**Questioner:** what color is her hair?
**Answerer:** dark brown
**Questioner:** what is her eye color?
**Answerer:** i can't tell, it's covered with white and black and red
**Questioner:** what is fence made of?
**Answerer:** wood
**Questioner:** what color is fence?
**Answerer:** it's metal fence with black metal fence
**Questioner:** is giraffe full grown?
**Answerer:** yes

**Image Caption:** *a number of people in a field with a kite in the air*
**Questioner:** how many people are there?
**Answerer:** i see 4
**Questioner:** what color is kite?
**Answerer:** it's hard to tell, but it looks black
**Questioner:** are they male or female?
**Answerer:** both are male
**Questioner:** what are they wearing?
**Answerer:** they are wearing shorts and t shirts
**Questioner:** what color are shorts?
**Answerer:** black

Figure 11: Example dialogues produced by the best performing ILQL agent on the Visual Dialogue "y/n" reward.

**example conservative y/n reward Visual Dialogues**

**Image Caption:** *Tour buses are lined up on the street waiting for people.*
**Questioner:** how many buses?
**Answerer:** 2
**Questioner:** what color are buses?
**Answerer:** white and red
**Questioner:** how many people?
**Answerer:** 2
**Questioner:** what gender are people?
**Answerer:** 1 is male and 1 is female
**Questioner:** what are they wearing?
**Answerer:** 1 is wearing shorts and other is wearing shorts and shirt
**Questioner:** what color is their hair?
**Answerer:** dark brown
**Questioner:** what color is their shirt?
**Answerer:** black
**Questioner:** what color is their pants?
**Answerer:** black
**Questioner:** what color is their shirt?
**Answerer:** black
**Questioner:** what color is their pants?
**Answerer:** black

**Image Caption:** *a living room with a table couch guitar and some pictures*
**Questioner:** what color is couch?
**Answerer:** brown
**Questioner:** what kind of pictures are on couch?
**Answerer:** 1 is UNK other is not sure
**Questioner:** what color is guitar?
**Answerer:** black
**Questioner:** what color is table?
**Answerer:** brown
**Questioner:** what color is table?
**Answerer:** brown
**Questioner:** what color is guitar?
**Answerer:** black
**Questioner:** what color are walls?
**Answerer:** white

**Image Caption:** *Little girl reaching through fence to pet a giraffe.*
**Questioner:** what is little girl wearing?
**Answerer:** she is wearing long sleeved shirt and jeans
**Questioner:** what color is fence?
**Answerer:** it's metal fence with black metal fence
**Questioner:** what is giraffe doing?
**Answerer:** it is just standing there

**Image Caption:** *Pictures of living room, dining room, kitchen and bedroom.*
**Questioner:** how many people are there?
**Answerer:** 2
**Questioner:** what are they doing?
**Answerer:** 1 is on top of other 1 is on other side of fridge
**Questioner:** what color is fridge?
**Answerer:** silver

Figure 12: Example dialogues produced by the best performing ILQL agent on the Visual Dialogue "conservative y/n" reward.

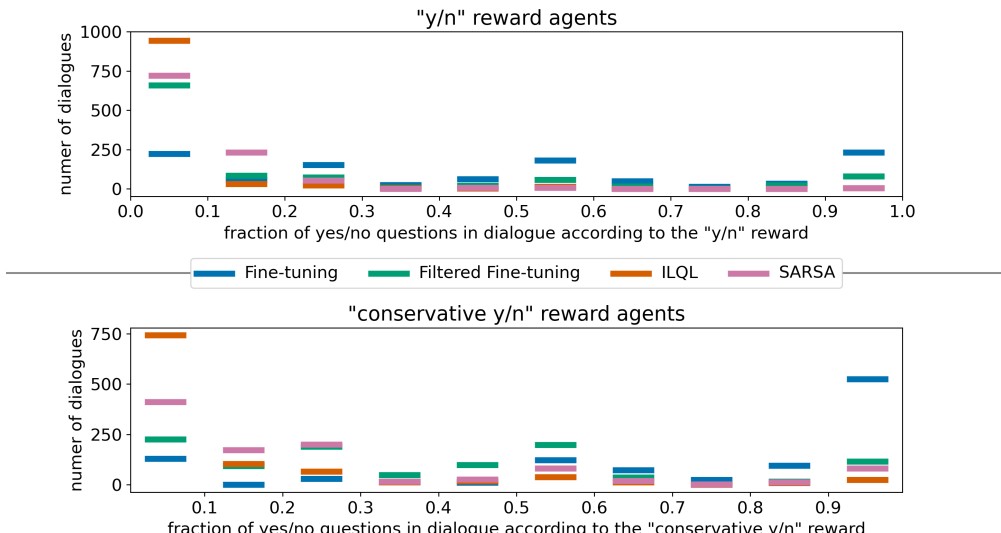

Figure 13: Top: Histogram of the fraction of yes/no questions asked per dialogue by Visual Dialogue agents trained on the "y/n" reward. Here yes/no questions are determined by the same exact match heuristic used by the "y/n" reward. ILQL agents ask fewer questions triggered as being yes/no than baselines. The best performing agent is used for all methods. Bottom: Histogram of the fraction of yes/no questions asked per dialogue by Visual Dialogue agents trained on the "conservative y/n" reward. Here yes/no questions are determined by the same exact match heuristic used by the "conservative y/n" reward. ILQL agents ask fewer questions triggered as being yes/no than baselines. The best performing agent is used for all methods.

| method | toxicity | noised toxicity |
|---|---|---|
| ILQL $\tau = 0.6$, $\beta = 1$ | -2.15±0.10 / 289.97±10.92 | -1.89±0.12 / 260.34±10.35 |
| ILQL $\tau = 0.6$, $\beta = 2$ | -1.16±0.08 / 222.18±9.68 | -0.99±0.09 / 202.09±8.99 |
| ILQL $\tau = 0.6$, $\beta = 4$ | -0.47±0.05 / 151.46±7.86 | -0.53±0.07 / 156.85±7.80 |
| ILQL $\tau = 0.6$, $\beta = 8$ | -0.12±0.02 / 79.20±5.38 | -0.23±0.05 / 73.12±4.88 |
| ILQL $\tau = 0.6$, $\beta = 16$ | -0.01±0.01 / 15.46±1.76 | -0.03±0.02 / 22.05±1.51 |
| ILQL $\tau = 0.6$, $\beta = 32$ | **0.00**±0.00 / 2.00±0.35 | **0.00**±0.00 / 9.24±0.16 |
| single-step RL $\beta = 1$ | -1.60±0.09 / 240.21 ± 10.10 | -1.72±0.12 / 233.17±9.89 |
| single-step RL $\beta = 2$ | -0.96±0.07 / 199.04±9.24 | -1.06±0.10 / 187.36±8.82 |
| single-step RL $\beta = 4$ | -0.34±0.04 / 111.13±6.76 | -0.41±0.06 / 108.76±6.53 |
| single-step RL $\beta = 8$ | -0.07±0.019 / 37.59±3.47 | -0.13±0.03 / 41.61±3.52 |
| single-step RL $\beta = 16$ | **0.00**±0.00 / 4.38±0.54 | **0.00**±0.0 / 1.09±0.38 |
| single-step RL $\beta = 32$ | **0.00**±0.00 / 0.03±0.01 | **0.00**±0.00 / 0.00±0.00 |
| Filtered Fine-tuning | **-0.74**±0.07 / 80.84±3.19 | **-1.61**±0.11 / 90.75±3.29 |
| Fine-tuning | **-3.51**±0.13 / 137.70±5.82 | **-3.48**±0.15 / 126.54±5.26 |

| method | upvotes real | upvotes model |
|---|---|---|
| ILQL $\tau = 0.6$, $\beta = 1$ | 6.29±0.15 / 39.11±1.99 | 8.38±0.12 / 25.15±1.43 |
| ILQL $\tau = 0.6$, $\beta = 2$ | 7.01±0.14 / 21.47±1.56 | 9.53±0.07 / 7.82±0.88 |
| ILQL $\tau = 0.6$, $\beta = 4$ | 7.47±0.14 / 7.91±0.63 | 9.99±0.01 / 1.38±0.09 |
| ILQL $\tau = 0.6$, $\beta = 8$ | 9.05±0.09 / 1.55±0.07 | **10.00**±0.00 / 0.78±0.02 |
| ILQL $\tau = 0.6$, $\beta = 16$ | 9.73±0.05 / 0.55±0.14 | **10.00**±0.00 / 0.54±0.02 |
| ILQL $\tau = 0.6$, $\beta = 32$ | **9.83**±0.04 / 0.18±0.02 | **10.00**±0.00 / 0.22±0.02 |
| single-step RL $\beta = 1$ | **6.23**±0.15 / 33.75±1.64 | 7.82±0.13 / 28.80±1.87 |
| single-step RL $\beta = 2$ | 4.66±0.16 / 15.81±0.82 | 8.96±0.10 / 9.34±0.64 |
| single-step RL $\beta = 4$ | 0.81±0.09 / 2.38±0.28 | 9.93±0.03 / 0.43±0.09 |
| single-step RL $\beta = 8$ | 0.09±0.03 / 0.00±0.00 | **10.00**±0.00 / 0.01±0.01 |
| single-step RL $\beta = 16$ | 0.09±0.03 / 0.00±0.00 | **10.00**±0.00 / 0.00±0.00 |
| single-step RL $\beta = 32$ | 0.09±0.03 / 0.00±0.00 | **10.00**±0.00 / 0.00±0.00 |
| Filtered Fine-tuning | **7.06**±0.14 / 100.77±4.15 | **7.86**±0.13 / 97.51±3.90 |
| Fine-tuning | **4.87**±0.16 / 127.65±5.48 | **4.87**±0.16 / 127.65±5.48 |

Table 13: All hyper-parameter settings evaluated across all Reddit Comment tasks, abalations, and baselines. The best performing setting for each baseline of abalation is bolded. The left result in each cell the the agent's average reward and the right result is the estimated entropy of the agent's policy, measured in nats. As $\beta$ is turned up, performance increases, but the entropy (or diversity) of the policy's outputs decreases. ILQL generally learns better performing and more diverse policies than baselines.

| method | score |
|---|---|
| ILQL $\tau = 0.7, \beta = 4$ | -2.23±0.03 |
| ILQL $\tau = 0.7, \beta = 8$ | -2.18±0.03 |
| ILQL $\tau = 0.7, \beta = 16$ | -2.18±0.03 |
| ILQL $\tau = 0.7, \beta = \infty$ | -6.00±0.00 |
| ILQL $\tau = 0.8, \beta = 4$ | **-2.13**±0.02 |
| ILQL $\tau = 0.8, \beta = 8$ | **-2.13**±0.03 |
| ILQL $\tau = 0.8, \beta = 16$ | -2.31±0.03 |
| ILQL $\tau = 0.8, \beta = \infty$ | -6.00±0.00 |
| ILQL $\tau = 0.9, \beta = 4$ | -2.30±0.03 |
| ILQL $\tau = 0.9, \beta = 8$ | -2.26±0.03 |
| ILQL $\tau = 0.9, \beta = 16$ | -2.36±0.03 |
| ILQL $\tau = 0.9, \beta = \infty$ | -6.00±0.00 |
| single-step RL $\beta = 4$ | -2.24±0.03 |
| single-step RL $\beta = 8$ | -2.26±0.03 |
| single-step RL $\beta = 16$ | **-2.23**±0.03 |
| single-step RL $\beta = \infty$ | -6.00±0.00 |
| 10% Filtered Fine-tuning | -2.93±0.06 |
| 30% Filtered Fine-tuning | -3.01±0.06 |
| 50% Filtered Fine-tuning | **-2.38**±0.03 |
| Fine-tuning | **-2.61**±0.03 |

Table 14: All hyper-parameter settings and baselines evaluated on the human Wordle dataset scraped from Tweets (see Section A.5). The best performing setting for each baseline of abalation is bolded. $\beta = \infty$ refers to greedily selecting actions with just the Q function. ILQL is generally better performing than baselines.

