# OpenReview forum: "Offline RL for Natural Language Generation with Implicit Language Q Learning"
_ICLR.cc/2023/Conference — ICLR 2023 poster_

### Official Review · Reviewer_LVpS · 2022-10-25

**Confidence:** 2
**Correctness:** 3
**Technical Novelty And Significance:** 3
**Empirical Novelty And Significance:** 3
**Recommendation:** 8

**Clarity, Quality, Novelty And Reproducibility:**

The paper is well written and introduces a new offline RL method tailored to language.

**Strength And Weaknesses:**

Strengths:
* The paper is well written and easy to follow.
* As reinforcement learning gains in popularity for NLP applications, this paper provides an important contribution by introducing scalable offline RL for language.
* The authors demonstrate that the method is easy to use and effective, and can handle existing data. In particular, it can learn to combine aspects of behaviors to achieve a higher reward than obtainable from the average of the data.
* The authors validate their method on a goal-directed question asking task (Visual Dialogue) and a Reddit comment generation task.

Weaknesses:
* ILQL requires a larger computational budget than supervised learning due to the additional cost of the behavioral model.

**Summary Of The Paper:**

The paper proposes Implicit Language Q-Learning (ILQL), a RL method, adapted from Implicit Q-learning, which can learn controlled language generation from a reward signal and a large-scale dataset of mixed quality in an offline fashion. The authors empirically validate their method on a dialogue task and reddit comment generation.

**Summary Of The Review:**

This paper introduces and validate a useful and effective new method for using reinforcement-learning to fine-tune large language models in a scalable offline fashion.

---

> ### Author Response · Authors · 2022-11-17
> **Response**
>
> Thank you for the positive review! We appreciate the positive feedback about the work and the constructive suggestions.

---

### Official Review · Reviewer_jx4F · 2022-10-26

**Confidence:** 3
**Correctness:** 4
**Technical Novelty And Significance:** 3
**Empirical Novelty And Significance:** 4
**Recommendation:** 8

**Clarity, Quality, Novelty And Reproducibility:**

Paper is clear, and authors have included a reproducibility statement which is more than sufficient. I appreciate inclusion of all the relevant hyperparameters and attached source code.

**Strength And Weaknesses:**

# Strengths

- Novel and sensible adaptation of a recent offline RL method, implicit Q learning, to the natural langauge generation setting, as well as various tricks that are needed to make learning work in this setting
- Thorough experimentation including toy examples as well as more realistic settings in visual dialog and reddit comment generation, with comparison to naive SARSA updates as well as full finetuning/finetuning only on high-reward trajectories.
- Thorough comparison to other baselines (when such baselines are included)---I appreciate authors' effort to sweep over hyperparameters and explain baselines in detail in Appendix---I think this is much more thorough than your typical ML paper.
- I like that the method clearly states limitations (e.g. longer training times compared to standard supervised RL, limitations to suboptimal data) in conclusion.

# Weaknesses

- I'm surprised that comparison to other offline RL baselines is deemed an "ablation" and only depicted in section 6.3/Table 3, and not throughout. In the main VisDial (table 1) and reddit results (table 2), ILQL is only compared to naive SARSA as well as FT/Filtered FT. Comparison to other offline RL methods used in the literature doesn't feel like an ablation to me - it feels like a fundamental comparison: is the added complexity required by ILQL worth the effort over other (perhaps conceptually simpler) offline RL algorithms? Moreover, how do the other offline RL algorithms handle reward shifts?

## Minor

- I find "Easy to Use" to be a fairly subjective/meaningless criterion in Figure 2.
- Page 6: Appendix ??
- It'd be great to explicitly point to Appendix A.4 in the paragraph "Ablations on choice of Offline RL algorithm".

**Summary Of The Paper:**

This paper proposes an offline RL method for natural language generation, ILQL, which extends a recently proposed offline RL method, Implicit Q Learning. The base method, Implicit Q Learning, uses expectile regression to learn a q function by limiting itself to only observed dataset actions. This paper translates the increased stability from this algorithm  to the NLP setting, while also presenting a few tricks required to making training work for NLG (e.g. pushing down weight of unseen tokens via a regularization term).

Authors first show a toy wordle example where the multi-step updates enabled by ILQL greatly improve performance in a contrived setting. They then evaluate on more realistic, open-ended visual dialog and reddit comments tasks, showing significant gains over standard finetuning, naive SARSA, and some other offline RL baselines. One of the particularly interesting experiments is that ILQL is robust to changes in the reward function (e.g. avoiding yes/no questions in visual dialog).

**Summary Of The Review:**

This paper clearly presents a novel offline RL method, ILQL, which shows strong results across a variety of diverse NLg tasks, especially in more adversarial settings (e.g. reward shift). Although comparison to offline RL baselines are not as thorough as the rest of the paper, I'm inclined to accept.

---

> ### Author Response · Authors · 2022-11-17
> **Response**
>
> Thank you for the positive review! We appreciate the positive feedback about the work and the constructive suggestions.
>
> We agree that putting the comparisons to other offline RL methods under “ablations” is an awkward choice. We've moved these into a separate section with a different heading. We also agree that a more thorough comparison to other offline RL methods under different reward functions would further improve the paper. Unfortunately, it’s difficult for us to complete these experiments during the rebuttal period due to some real-life constraints and compute limitations, but we will aim to add these in the final version of the paper. We agree that this would strengthen the results.

---

### Official Review · Reviewer_8Evy · 2022-10-28

**Confidence:** 4
**Correctness:** 1
**Technical Novelty And Significance:** 3
**Empirical Novelty And Significance:** 3
**Recommendation:** 3

**Clarity, Quality, Novelty And Reproducibility:**

There is one broken hyperref link on page 6 that refers to an Appendix section.

Other than that, the algorithm seems generally novel. The quality and clarity seem ok.

**Strength And Weaknesses:**

Strength:
- The paper proposes a novel learning paradigm ILQL (Figure 3). The idea of directly "adding" advantage Q-V to p(vocab) is very appealing.
- The paper also re-configured the CQL pessimism penalty and re-adapted it to work with discrete action space, which is novel.
- The experimental result seems strong to convince me this algorithm works on a wide range of task domains.

Weakness:
- Section 5 is wrong, which affected all the follow-up experimental comparisons between ILQL and "SARSA". There are a few reasons why this is wrong (or "trivial"):
   1. I don't think the author understands the concept of SARSA vs. Q-learning (ILQL). SARSA, first of all, is an on-policy learning algorithm, while Q-learning (ILQL) is an off-policy learning algorithm. In the offline RL setting, an on-policy learning algorithm would never make any sense because on-policy exploration is forbidden.
   2. SARSA provides a convergence guarantee under two specific conditions: 1). Infinite visits to every state-action pair; 2). The learning policy becomes greedy in the limit. Offline RL setting (and Figure 4 worked example) satisfies NONE of these conditions. If we read the text in the paragraph, the learning policy $\pi_{\text{upper bound}}$ was never updated (it stays fixed). The suboptimal result is trivial and obvious. I encourage the author to read [1]. Comparing Q-learning with SARSA in an offline RL setting makes no sense.
   3. I think this paper tries to follow the original IQL too closely and re-iterate the point made in IQL but without enough finesse and care. The original IQL paper never suggested that SARSA would ever work in an offline RL setting. Also, IQL > SARSA is already shown in the original IQL paper; what is the point of showing this again?
   4. Overall, I think Section 5 is largely misleading to people who are not familiar with RL and particularly Offline RL.

Suggestion:
- The equation on page 5, $L^c_{Q, V}(\theta)$, the author should be able to offer a short proof (of a few lines) that show it's equivalent to CQL pessimism penalty. This would make the statement in the paragraph more concrete: "this CQL loss term is no more expensive than, and in fact equivalent to, a standard cross-entropy loss at the token level".
- I think the authors should consider re-orient this paper to be more NLP-focused and drop the comparison to SARSA in this offline setting. The results aren't surprising to any RL researchers and only serve to confuse them. The proposed algorithm ILQL is novel enough! It doesn't need the motivation to do better than SARSA.
- I think the authors should rethink the pitch and conduct experiments that make sense to the pitch.

References:

[1] Convergence Results for Single-Step On-Policy Reinforcement-Learning Algorithms (https://link.springer.com/content/pdf/10.1023/A:1007678930559.pdf)

**Summary Of The Paper:**

This paper investigates IQL, an offline RL training algorithm, applied to training a language model to generate a sequence that will maximize a utility function (such as a user reward model).

**Summary Of The Review:**

The paper is generally interesting, and IQIL is sufficiently different from other Offline RL + LM fine-tuning/training papers. However, I don't think the main claim (IQIL/Q-learning > SARSA) in an offline RL setting makes sense. Though this paper has so many interesting ideas, great experimental results, and on a diverse collection of datasets, I cannot recommend an accept at this point.

---

> ### Author Response · Authors · 2022-11-10
> **Response to Review**
>
> Thank you for the detailed review. We’re glad to hear that you found the work to be interesting and the results convincing. It appears as though the main reason that you recommend rejecting the work is the inclusion of the comparison to SARSA and the discussion in Section 5. We believe there is a misunderstanding here, perhaps due to the choice of phrasing in Section 5: we are not claiming that SARSA is an effective (or even correct) strategy for solving off-policy RL problems – as you correctly point out, it is not a valid approach for learning optimal policies, since it only learns the value function of the behavior policy. However, as we elaborate below, this approach is commonly used in the literature both in natural language processing (often called a “reward model” [1, 2, 3]) and in offline RL (e.g., “one step RL” and other related methods [3, 4]). We’ve revised this section to clarify this point. We would also emphasize that the main claim of the paper is not that ILQL outperforms SARSA, but that the ILQL offline RL method is an effective strategy for NLP problems (see introduction, Section 1: “Our main contribution is…”). That is, the pitch is already NLP-focused, just as you suggest. We would appreciate it if you would let us know whether these revisions address your concern, or if there is another modification that you would prefer.
>
> **Section 5, SARSA, and offline RL:** Our paper proposes a value-based offline RL method that performs dynamic programming for goal-directed NLP tasks. Our claim is that such an approach is effective for a variety of NLP problems, as shown in our experiments. Prior works in NLP have proposed alternative approaches that do not perform full dynamic programming (see, e.g., [1, 2, 3]). Such methods learn “reward models” or value functions that essentially estimate the value of the behavior policy, often via Monte Carlo regression. The value function of the behavior policy can be learned via Monte Carlo, SARSA, etc., but the result in these cases is the same and, as you correctly point out, is clearly suboptimal. Such approaches have also been proposed in the offline RL literature [4, 5], with the motivation that, though they are in theory suboptimal, they can be simple and effective in practice. Therefore, it seemed reasonable for us to include a discussion of why full dynamic programming approaches, such as ILQL, should be preferred over these techniques. We have clarified this section to add more context to this discussion. We also renamed “SARSA” to “single-step RL” to better reflect the terminology used in the prior offline RL literature [4] and avoid misunderstandings. We would appreciate any further suggestions about how this section could avoid misleading the reader. We of course agree with you that SARSA is not a particularly logical approach to apply to off-policy RL problems, and we hope the revisions make this clear, but we contrast our method to filtered BC, SARSA, and other methods that do aim to solve for the optimal value function precisely because such methods have been used in the literature and form a natural point of reference. Your criticism of these prior methods is fair, but we believe it’s important for our paper to discuss these issues precisely to avoid the kinds of misunderstandings that your review cautions against (and we would appreciate any suggestions for how to do this better!).
>
> That said, we do not believe that anything currently in Section 5 is incorrect, though we would certainly appreciate the reviewer pointing out any specific incorrect statements!
>
> **Regarding the pitch** (“I think the authors should rethink the pitch”): We believe the paper already is oriented as addressing applications in NLP, and we strongly agree with you that this is the right way to present it! Are there any specific sections you would like us to revise to clarify this, or any parts that inadvertently created a different impression?
>
> As a general point regarding the review, we are glad that you see the work as novel and valuable, and we are saddened to see the reason for rejection concerns primarily our choice of comparison. We hope that we can address this issue by revising the paper, and we would appreciate a response as to whether our changes and clarifications above address this problem.
>
> [1] Young, Tom, et al. "Recent trends in deep learning based natural language processing."
>
> [2] Gu, Jiatao, et al. "Learning to translate in real-time with neural machine translation."
>
> [3] Su, Pei-Hao, et al. "On-line active reward learning for policy optimisation in spoken dialogue systems."
>
> [4] Brandfonbrener, David, et al. "Offline rl without off-policy evaluation."
>
> [5] Chen, Lili, et al. "Decision transformer: Reinforcement learning via sequence modeling."

---

> > ### Comment · Reviewer_8Evy · 2022-11-17
> > **Response**
> >
> > When I read the updated paper, the only change (in blue) that I can see is:
> >
> > > Such methods have been referred
> > to in the NLP literature as “reward models" (Young et al., 2017; Gu et al., 2016; Su et al., 2016) and
> > in the offline RL literature “one step RL" or SARSA (Brandfonbrener et al., 2021; Chen et al., 2021).
> > While in principle such methods should not lead to optimal policies, in practice they often constitute
> > an appealing approximation due to their ease of use.
> >
> > There are two problems. First, Chen et al., 2021, the Decision Transformer is not one-step RL nor SARSA; it's BC. Writing sentences like this is, again, very misleading to readers who are unfamiliar with RL (which I argue is the target audience of this paper, aka NLP folks). Second, I think Section 5, which has the title: "PROOF OF CONCEPT," the reasoning is slightly off. Let me unpack it for you:
> > 1. One-step RL is empirically useful in offline RL. We have a reasonable suspicion it might work.
> > 2. We show in a constructed toy example that one-step RL fails (but here, the reason why it doesn't work and the reason why it works in empirical offline RL are perpendiculars -- it works for offline RL due to policy shift; it doesn't work here due to un-updated sampling policy) (Are you claiming policy shift is not a big problem in your offline RL domain?)
> > 3. For empirical results, we show that it doesn't work.
> >
> > I think building your paper's core insight on one-step RL is a mistake. I'm also curious why one-step RL and CQL work well on other offline RL datasets but fail to work for language tasks. Maybe this is what your toy example is trying to illustrate, but again, you are hitting a different issue with the toy example (see above). Is there anything inherently different about the dataset you are trying on, so we really should do multi-step backups instead of single-step backups? Also, since BC algorithms (DT) don't work well either, there does seem to be some evidence that the NLP domains are special. It will be a nice addition to the paper if you have the time and energy to investigate it.

---

> > > ### Author Response · Authors · 2022-11-17
> > > **Further revisions**
> > >
> > > Thank you for the additional feedback. I've revised the draft again to address this.
> > >
> > > I'll discuss the specific changes below. But first, a general point: I think there is some major misunderstanding regarding the importance of Section 5 in the paper. I want to go through this very carefully to make sure we address your concern, which I think stems from either a misunderstanding or some poor communication on our part.
> > >
> > > It is definitely not the intention to "build the paper's core insight on one-step RL." **Is there any language in the current paper that suggests this?** If so, we should definitely change it! I've read over the intro again multiple times, and I can't find anything there that I imagine would create the impression that the contrast to one-step RL is a central part of the paper's **insight**. As we wrote before, the inclusion of Section 5 is meant to address a very specific issue: one-step methods, BC-based methods, "reward models," and other such algorithms have been used widely in this community, as the modified (highlighted in blue) text clarifies. Therefore, it seemed important to address this **somewhere** in the paper. If you strongly believe this is inappropriate, we can of course remove it, but I find this puzzling -- I agree with you that such methods should not be great offline RL methods, and I want the paper to make this point clearly and correctly. That's the whole point of that section!
> > >
> > > **In summary**, please tell us if there is anything outside of Section 5 that (unintentionally) creates the impression that the one-step comparison is a central part of the paper's contribution or the paper's insight, so that we can fix this.
> > >
> > > > First, Chen et al., 2021, the Decision Transformer is not one-step RL nor SARSA; it's BC.
> > >
> > > Yes, agreed, this citation was misplaced! I revised this paragraph to add: "Some works also proposed behavioral cloning methods that filter the training data or use conditioning to clone high-reward trajectories~\citep{https://doi.org/10.48550/arxiv.2106.01345} -- though these methods are based on different principles, they also employ Monte Carlo estimates of the cumulative reward in place of value functions learned with dynamic programming." [see section highlighted in blue in Sec 5] **Does this fully address this issue**?
> > >
> > > > Second, I think Section 5, which has the title: "PROOF OF CONCEPT," the reasoning is slightly off.
> > >
> > > I would like to address this, but I'm having trouble understanding the concern precisely. What I did in this revision is add some text in the "Synthetic Wordle Task" paragraph (highlighted in blue) that explicitly states that we are **not** implying that the thing that makes this example fail is the same as what accounts for the results in Section 6, to address your point that "the reason why it doesn't work and the reason why it works in empirical offline RL are perpendiculars" (which is true). But to get into the details more:
> > >
> > > > 1. One-step RL is empirically useful in offline RL. We have a reasonable suspicion it might work.
> > > > 2. We show in a constructed toy example that one-step RL fails (but here, the reason why it doesn't work and the reason why it works in empirical offline RL are perpendiculars -- it works for offline RL due to policy shift; it doesn't work here due to un-updated sampling policy (Are you claiming policy shift is not a big problem in your offline RL domain?)
> > > > 3.For empirical results, we show that it doesn't work.
> > >
> > > That all seems right to me. Hopefully the additional clarifications make this clearer. We are not claiming anything about why it doesn't work for the empirical offline RL experiments, because we don't actually know. I agree that studying this would be really interesting, but I don't think we can pull it off in the two remaining days of the rebuttal period.
> > >
> > > The updated text hopefully also makes it clearer that the Wordle example is intended to evaluate this aspect of language offline RL methods -- whether this aspect is critical for real-world results or not, it seems important to have a proper dynamic programming algorithm rather than a one-step/SARSA/MC/BC method (for the reasons stated in your original review!), so it seems good to have a benchmark to test it. Note that the task here is **not** the toy MDP, but a dataset with actual text for the Wordle game.

---

> ### Author Response · Authors · 2022-11-15
> **Response**
>
> Thank you again for the review, we wanted to check if you've had a chance to go over our responses and revisions, and whether these changes address the issues you've raised. We believe these issues should be quite fixable, but it would be helpful for us to get your feedback here to make sure we understood your concerns correctly.

---

### Decision · Program_Chairs · 2023-01-20

**Decision:**

Accept: poster

**Justification For Why Not Higher Score:**

* More thorough comparison to other offline RL methods is needed, which was not addressed in rebuttal.
* Reviewer had concern on the discussion with SARSA as an online RL method, and with other offline RL methods

**Justification For Why Not Lower Score:**

Reviewers found the algorithm sound and new, and the performance impressive.

**Metareview: Summary, Strengths And Weaknesses:**

The paper proposes an offline RL algorithm, ILQL, for natural language generation. The approach is an extension of the recent Implicit Q Learning with a few new techniques (e.g., additional regularizations) designed for text tasks. Experiments on open-ended visual dialog and reddit comments show good improvement over standard finetuning, naive online RL (SARSA) and certain offline RL baselines. The reviewers have suggested more thorough comparison to other offline RL methods under different reward functions. The paper also needs to make clearer the comparison with SARSA as an online RL method.

**Note From Pc:**

if the above contains the word "oral" or "spotlight" please see: "oral" presentation means -> notable-top-5% and "spotlight" means -> notable-top-25%. As stated in our emails, we are disassociating presentation type from AC recommendations